# Herbivorous insects independently evolved salivary effectors to regulate plant immunity by destabilizing the malectin-LRR RLP NtRLP4

Xin Wang[1,2], Jia-Bao Lu[1], Yi-Zhe Wang[1], Xu-Hong Zhou[3], Jian-Ping Chen[1,2], Chuan-Xi Zhang[1], Jun-Min Li[1]*, Hai-Jian Huang[1]*

[1]State Key Laboratory for Quality and Safety of Agro-Products, Key Laboratory of Biotechnology in Plant Protection of MARA, Zhejiang Key Laboratory of Green Plant Protection, Institute of Plant Virology, Ningbo University, Ningbo, China; [2]Institute of Applied Ecology, Fujian Agriculture and Forestry University, Fuzhou, China; [3]Office of Science and Technology, Yunnan University of Chinese Medicine, Kunming, China

*For correspondence:
lijunmin@nbu.edu.cn (J-ML);
huanghaijian@nbu.edu.cn (H-JH)

Competing interest: The authors declare that no competing interests exist.

## eLife Assessment

This study provides an **important** contribution by showing that whiteflies and planthoppers use salivary effectors to suppress plant immunity through the receptor-like protein RLP4, suggesting convergent evolution in these insect lineages. The topic is of clear interest for understanding plant-insect interactions and offers ideas that could stimulate further research in the field. The authors provide **convincing** evidence for the functional roles of the salivary effectors.

**Abstract** Plants utilize receptor-like proteins and receptor-like kinases (RLPs/RLKs) to perceive and respond to a wide variety of invading pathogens and insect herbivores. While the strategies employed by microbial pathogens to suppress plant immunity have been well characterized, it remains unclear how herbivorous insects counteract receptor-mediated defenses. Here, we show that salivary effectors evolve independently in whiteflies and planthoppers to dampen RLP4-mediated plant immunity. RLP4, as a leucine-rich repeat RLP (LRR-RLP), confers plant resistance against herbivorous insects by forming the RLP4/SOBIR1 complexes. In the whitefly *Bemisia tabaci*, BtRDP, the Aleyrodidae-specific salivary sheath protein, interacts with RLP4 from multiple plant species and promotes its ubiquitin-dependent degradation. Overexpression of NtRLP4 in transgenic plants exerts a detrimental effect on *B. tabaci* by exploiting the crosstalk between the salicylic acid and jasmonic acid pathways. Conversely, overexpression of BtRDP or silencing of NtRLP4 effectively alleviates such negative effects. In planthopper *Nilaparvata lugens*, the Delphacidae-restricted salivary protein NlSP104 also targets and promotes the degradation of OsRLP4 from rice plants. These findings reveal convergent evolution of salivary proteins in insects and underscore the complex interactions between plants and herbivorous insects.

## Introduction

Plants have evolved sophisticated defense systems to withstand continuous threats from microbial pathogens and herbivorous insects. These defenses function at multiple levels, encompassing physical barriers such as the cuticle and cell wall, chemical defenses that include toxic secondary

**eLife digest** Plants cannot escape from insects, so they rely on their own defense systems. One key strategy involves proteins on the cell surface that act as sensors. These sensors detect insect attacks and trigger protective responses within the plant.

Scientists have long known that microbes can disable these sensors, thereby weakening plant defences. However, it has been unclear whether plant-eating insects use similar tactics. Many insects feed by inserting needle-like mouthparts (stylets) into plants and releasing saliva, which contains proteins capable of altering plant responses.

To investigate this, Wang et al. studied two major crop pests: the whitefly *Bemisia tabaci* and the brown planthopper *Nilaparvata lugens*. They focused on a plant sensor called RLP4, a surface protein that helps plants recognize insect attack and activate defenses. The researchers found that both insects produce salivary proteins that bind to RLP4 and trigger its breakdown inside plant cells. This weakens the plant's defenses and makes feeding easier for the insects.

Experiments in tobacco and rice plants showed that increasing RLP4 levels improved resistance to these pests. In contrast, reducing RLP4 levels or introducing the insect salivary proteins made plants more susceptible. Although the two insect proteins are unrelated, they perform the same function, suggesting that different insects have independently evolved similar strategies to overcome plant defenses.

These findings reveal a shared mechanism used by plant-eating insects and provide new insight into plant–insect interactions. In the future, this knowledge could help guide the development of crops with improved resistance to insect pests. However, further research is needed to determine how widespread this mechanism is and how it can be effectively applied in agriculture.

metabolites and anti-nutritive compounds, and indirect defenses involving the emission of herbivore-induced volatiles to attract natural enemies of the herbivores (*Howe and Jander, 2008*; *Wang et al., 2024*). Beyond these general strategies, plants also rely on the highly specialized molecular immune responses to detect and rapidly respond to invaders. The activation of innate immunity by membrane-localized receptors is a highly conserved mechanism among eukaryotes (*Nürnberger et al., 2004*). In sessile plants, the first line of defense against potential pathogenic microbes or insects relies on cell surface pattern-recognition receptors (PRRs) (*Macho and Zipfel, 2014*). The plant genome can encode hundreds of PRRs, including receptor-like kinases (RLKs) and receptor-like proteins (RLPs) (*Macho and Zipfel, 2014*). RLKs consist of different types of N-terminal ectodomains, a single trans-membrane domain, and a cytoplasmic protein kinase domain, while RLPs possess a similar structure but lack the kinase domain (*Zipfel, 2014*; *Gust and Felix, 2014*). As RLPs lack the intracellular signaling domains, they are anticipated to associate with adaptor kinases to form the bimolecular receptor kinases. For example, suppressor of BAK1-interacting receptor-like kinase 1 (SOBIR1) is reported to act as a common adaptor for most, if not all, of the leucine-rich repeat RLP (LRR-RLP) (*Gust and Felix, 2014*). Despite the large number of PRRs encoded by plant genomes, only a few of them have been well characterized for their functions in recognizing microbial-associated molecular patterns (MAMPs), damage-associated molecular patterns (DAMPs), and herbivore-associated molecular patterns (HAMPs) (*Zipfel, 2014*; *Steinbrenner et al., 2020*). For instance, FLS2 is known to recognize bacterial flagellin (*Chinchilla et al., 2006*), EFR can recognize the bacterial elongation factor Tu (*Zipfel et al., 2006*), and EIX2 is responsible for the recognition of the fungal ethylene-inducing xylanase (*Ron and Avni, 2004*).

Piercing–sucking insects, such as whiteflies, aphids, and planthoppers, can damage plants by feeding or transmitting viruses. They inject a mixture of saliva into plant tissues, which enhances their feeding efficiency by binding to leaked calcium, degrading extracellular DNA, and regulating plant hormonal signals (*Tian et al., 2021*; *Huang et al., 2019*; *Chen et al., 2019*; *Xu et al., 2019*). In response, plants have evolved the ability to detect herbivores by sensing HAMPs present in insect saliva (*Steinbrenner et al., 2020*; *Chen and Mao, 2020*). This triggers pattern-triggered immunity (PTI), including mitogen-activated protein kinase (MAPK) cascades, reactive oxygen species (ROS), and hormone signaling (*Guiguet et al., 2016*; *Wu and Baldwin, 2009*; *Jiang et al., 2019*). A few salivary proteins have been reported to activate plant PTI, but the specific PRRs that recognize these

proteins and trigger plant immunity remain largely unexplored (*Gao et al., 2023*; *Guo et al., 2020*; *Bos et al., 2010*). One exception is the inceptin receptor (INR), which is a legume-specific LRR-RLP that can recognize the Vu-In in insect oral secretions (*Steinbrenner et al., 2020*). Moreover, plant PRRs have been extensively reported to be targeted by pathogen effectors (*Zhang and Zeng, 2020*; *Huang et al., 2020*; *Shan et al., 2008*). However, it remains largely unknown how insects cope with plant PRRs.

The whitefly *Bemisia tabaci* (Hemiptera: Aleyrodidae) and brown planthopper *Nilaparvata lugens* (Hemiptera: Delphacidae) are notorious pests globally. During their feeding process, both insects secrete abundant species-specific salivary proteins into host plants (*Huang et al., 2016*; *Huang et al., 2021*). It is reported that plants initiate defense responses upon the recognition of insect feeding (*Gao et al., 2023*; *Gao et al., 2022*). However, the specific PRRs responsible for this process are still unknown. Both insects can successfully ingest phloem saps with limited plant defenses. At present, it remains unknown how insects attenuate plant defenses. Given that numerous defense mechanisms are conserved across plants (*Nürnberger et al., 2004*; *Mukhtar et al., 2011*), it is interesting to investigate whether insects evolve a similar strategy to aid feeding during convergent evolution. In this study, the functions of RLPs in the plant immunity responsive to piercing–sucking insects were investigated, and *B. tabaci* and *N. lugens* were demonstrated to independently evolve salivary proteins to regulate the post-translational modification of RLPs.

## Results

### BtRDP is most abundantly expressed in salivary glands and secreted into host plants

Based on the transcriptomic analysis, BtRDP (*B. tabaci* RLP-degrading protein; GenBank accession: MN738093) was identified as the most abundantly expressed gene in salivary glands (*Huang et al., 2021*). It was a secretory protein featuring an N-terminal signal peptide at the N-terminal, and six unique peptides from BtRDP were detected in the *B. tabaci* watery saliva (*Figure 1—figure supplement 1*). As there was no gene homolog identified in the NCBI nr database, BtRDP was subjected to BLAST analysis against the transcriptomic or genomic data of 30 insect species. RDP was distributed in all the analyzed Aleyrodidae species, while no RDP homolog was detected in other species (*Supplementary file 1A*). Phylogenetic analysis indicated that RDPs from five *B. tabaci* cryptic species were clustered in the same clad (*Figure 1—figure supplement 2*).

A group of *B. tabaci* was allowed to feed on *Nicotiana tabacum* plants for 24 hr, and the presence of BtRDP in the infested and non-infested samples was assayed. One band between 26 and 33 kDa was detected in the *N. tabacum* plants infested by *B. tabaci*, but not in the non-infested plants (*Figure 1a*). This band matched the molecular weight of BtRDP identified in the salivary glands, suggesting BtRDP as a salivary protein. According to western blotting, qPCR, and transcriptomic analysis, BtRDP was nearly exclusively expressed in the salivary glands (*Figure 1b, c*; *Figure 1—figure supplement 3*; *Figure 1—figure supplement 4*). In addition, BtRDP was highly expressed in the adult stages, but lowly expressed in the egg, nymph, and pseudopupa stages (*Figure 1d*; *Figure 1—figure supplement 3*). According to immunohistochemical (IHC) staining, BtRDP was specifically distributed in the principal salivary gland, but not in the accessory salivary gland or other tissues (*Figure 1e*). For most piercing–sucking insects, two types of saliva (gel and watery saliva) are secreted into plant tissues. The salivary sheath, formed from gel saliva, serves as an insoluble lining along the stylet path, potentially providing a scaffold for effector delivery (*Huang et al., 2023*). The salivary sheath secreted by *B. tabaci* was also stained with the anti-BtRDP serum. The BtRDP signal was mainly detected in the protuberant structure of sheath (*Figure 1f*), suggesting that BtRDP attaches to salivary sheath after secretion.

### BtRDP is important for *B. tabaci* feeding on tobacco plants

RNA interference was conducted to investigate the role of BtRDP in whitefly feeding. Treatment with ds*BtRDP* efficiently and specifically suppressed the target gene at both the transcript and protein levels (*Figure 2a*; *Figure 2—figure supplement 2*). There was no significant difference in the salivary sheath morphology between ds*BtRDP*-treated and ds*GFP*-treated *B. tabaci* (*Figure 2—figure supplement 2*). However, the salivary sheath length of ds*BtRDP*-treated *B. tabaci* was significantly

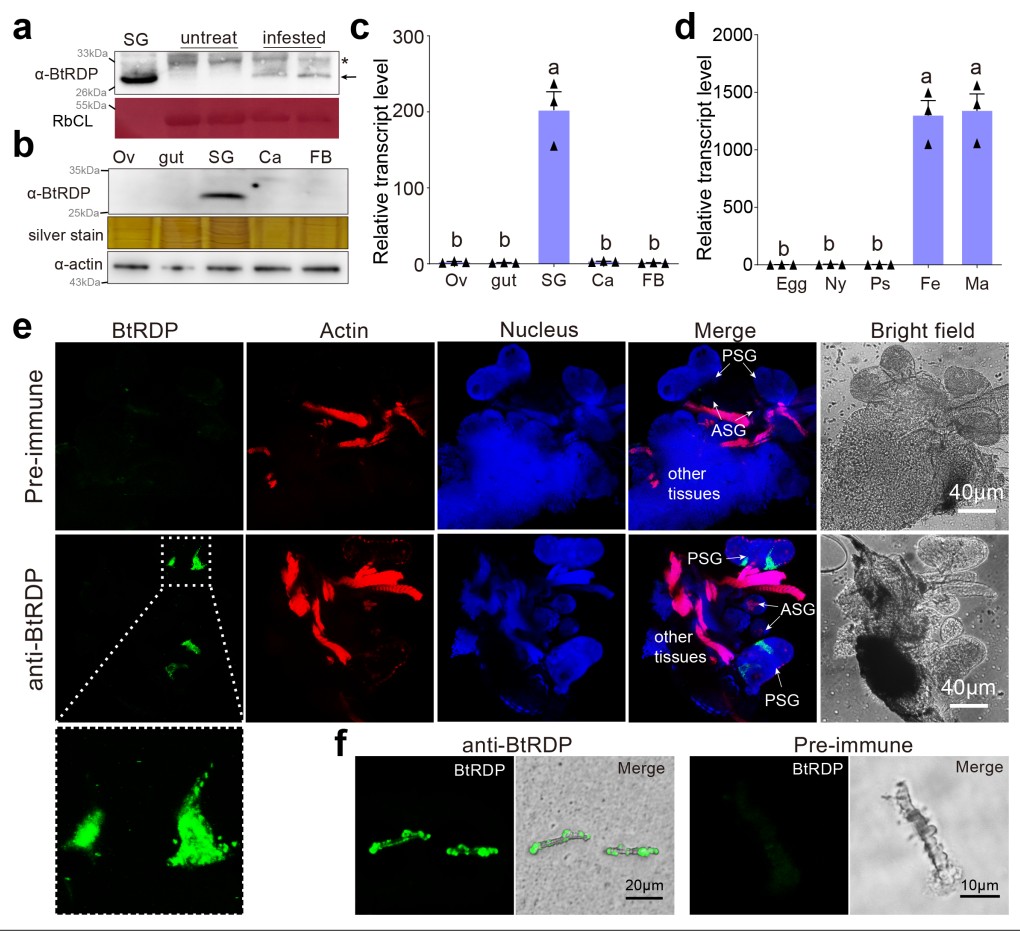

**Figure 1.** BtRDP is a salivary protein and secreted into plants. (**a**) Detection of BtRDP in *Nicotiana tabacum* plants. The untreated and *Bemisia tabaci*-infested tobacco plants, as well as the salivary gland samples, are collected for western blotting assays. Asterisk indicates non-specific binding. Rubisco staining (RbcL) is conducted to visualize the amount of sample loading. (**b, c**) Detection of BtRDP in different tissues by western blotting and quantitative real-time PCR (qRT-PCR) assays. Ov, ovary; SG, salivary gland; Ca, carcass; FB, fat body. Silver staining and anti-actin serum are used to visualize the sample loading. (**d**) Expression patterns of BtRDP in different development stages. Ny, nymph; Ps, pseudopupa; Fe, female; Ma, male. *B. tabaci tubulin* is used as an internal control. The relative quantitative method ($2^{-\Delta\Delta Ct}$) is used to evaluate the quantitative variation. Data are presented as mean values ± SEM ($n$ = 3 independent biological replicates). Different lowercase letters indicate statistically significant differences at the $p < 0.05$ level according to one-way ANOVA test followed by Tukey's multiple comparisons test. Immunohistochemical staining of BtRDP in salivary glands (**e**) and salivary sheath secreted from *B. tabaci* (**f**). The salivary gland and its nearby tissues are dissected and incubated with anti-BtRDP serum or pre-immune serum conjugated with Alexa Fluor 488 NHS Ester (green) and actin dye phalloidin–rhodamine (red). The nucleus is stained with 4′,6-diamidino-2-phenylindole (DAPI, blue). PSG, principal salivary gland; ASG, accessory salivary gland. The lower images represent the enlarged images of the boxed area in the upper image. Experiments are repeated thrice for (**a, b**), while twice for (**e, f**). Similar results are observed, and representative images are displayed.

The online version of this article includes the following source data and figure supplement(s) for figure 1:

**Source data 1.** PowerPoint file containing original western blots for *Figure 1a*, indicating the relevant bands and treatments.

**Source data 2.** Original files for western blot analysis displayed in *Figure 1a*.

**Source data 3.** PowerPoint file containing original western blots for *Figure 1b*, indicating the relevant bands and treatments.

**Source data 4.** Original files for western blot analysis displayed in *Figure 1b*.

**Figure supplement 1.** Characteristic of BtRDP.

*Figure 1 continued on next page*

*Figure 1 continued*

**Figure supplement 2.** Analysis of insect RDP.

**Figure supplement 3.** Expression patterns of *Bemisia tabaci BtRDP*.

**Figure supplement 4.** Expression patterns of *BtActin* and *Bt18s rRNA*.

shorter than that of the ds*GFP*-treated control (***Figure 2—figure supplement 2***). Furthermore, treatment with ds*BtRDP* did not affect the survivorship of whiteflies (***Figure 2—figure supplement 2***). A reproduction analysis was conducted, which indicated that the ds*BtRDP*-treated whiteflies oviposited fewer eggs than the control (***Figure 2b***). The feeding behavior of whiteflies was monitored using the electrical penetration graph (EPG) technique, which distinguishes nonpenetration (np), pathway (C), phloem salivation (E1), and phloem ingestion (E2) phases (***Lu et al., 2021***; ***Figure 2—figure supplement 3***). Compared to the ds*GFP* control, ds*BtRDP*-treated *B. tabaci* exhibited a significant reduction in phloem ingestion and an extended pathway duration, indicating that BtRDP is necessary for efficient feeding (***Figure 2c***).

Salivary proteins are reported to function in intercellular space and/or extracellular space after being secreted into host plants (***Huang et al., 2023***; ***Yan et al., 2023***). To investigate the role of BtRDP in different subcellular locations of host plants, we constructed two transgenic *N. tabacum* lines overexpressing BtRDP: one carrying the full-length coding sequence that included the signal peptide (oeBtRDP), which is expected to be secreted into the apoplast (extracellular space), and the other one lacking the signal peptide (oeBtRDP⁻ˢᵖ), which is likely to be retained in the cytoplasm. Transgenic plants containing the empty vector (EV), which exerted no significant influence on insect performance compared with the WT (***Figure 2—figure supplement 4***), were used as negative controls. Fecundity and two-choice assays demonstrated that BtRDP, whether localized in the apoplast (oeBtRDP) or cytoplasm (oeBtRDP⁻ˢᵖ), enhanced whitefly settling and oviposition relative to the EV controls (***Figure 2d–i***; ***Figure 2—figure supplement 5***), indicating that BtRDP promotes whitefly feeding behavior irrespective of its subcellular location.

## BtRDP interacts with the NtRLP4/NtSORBIR1 complex

To investigate the underlying mechanism of BtRDP in improving whitefly feeding, yeast two-hybrid (Y2H) screening against a *N. benthamiana* cDNA library was conducted using BtRDP⁻ˢᵖ as a bait. Totally, seven proteins were found to potentially interact with BtRDP⁻ˢᵖ (***Supplementary file 1B***), including an NbRLP4 (accession Niben261Chr07g1310001.1 in Niben261 genome). NbRLP4 possesses a predicted N-terminal signal peptide, a malectin-like domain, an LRR domain, and a transmembrane domain. NbRLP4 and its closest homology in *N. tabacum* (NtRLP4; GenBank accession: XM_016601109) did not contain the kinase domain (***Figure 3a***).

The relative expression of *NtRLP4* in response to *B. tabaci* infestation was investigated. Based on quantitative real-time PCR (qRT-PCR) results, the transcript level of *NtRLP4* was not significantly influenced by *B. tabaci* (***Figure 3—figure supplement 1***). However, a decrease in the protein level of NtRLP4 was detected by western blotting assay (***Figure 3b***). Pairwise Y2H assays revealed that NtRLP4$_{(23–541)}$ (a truncated version lacking the signal peptide and transmembrane domains) interacted with BtRDP⁻ˢᵖ (***Figure 3c***). To investigate the specificity of the NtRLP4–BtRDP interaction, another RLP protein NtCf9 with the highest sequence similarity to tomato resistance protein Cf9 (***Figure 3—figure supplement 2***; ***Jones et al., 1994***), as well as three randomly selected salivary proteins from *B. tabaci* (BtFTSP, QHB15613; BtSP16.3, MN738005; BtSP37.4, MN738158), were used as controls. As a result, BtRDP⁻ˢᵖ did not interact with the truncated NtCf9 construct, in which the signal peptide and transmembrane domains were removed (***Figure 3c***). Also, NtRLP4$_{(23–541)}$ could not interact with the other three salivary proteins without signal peptides (***Figure 3—figure supplement 3***). These findings raise the question of which domain of NtRLP4 is responsible for binding to BtRDP, as identifying the interacting domain can help to infer where the salivary protein contacts the receptor in plants. We therefore dissected the NtRLP4 domains accordingly. The results showed that it was the LRR domain, rather than the malectin-like domain or the short intercellular region, that was required for the NtRLP4-BtRDP interaction (***Figure 3c***). To investigate the shortest LRR sequence responsible for BtRDP interaction, the LRR domain was divided into three segments. However, none of these segments interacted with BtRDP⁻ˢᵖ (***Figure 3—figure supplement 3***). Co-immunoprecipitation (Co-IP)

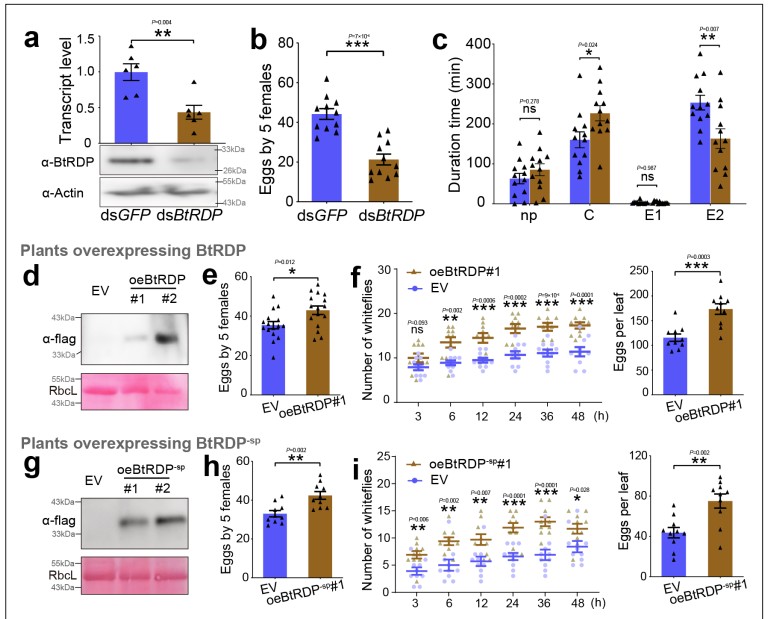

**Figure 2.** Effects of BtRDP on *Bemisia tabaci*. (**a**) Treating *B. tabaci* with ds*BtRDP* significantly reduces the transcript and protein level of target gene. The ds*GFP*-treated *B. tabaci* is used as a control. Effects of *BtRDP* knockdown on insect reproduction (**b**) and feeding behavior (**c**). Electrical penetration graph (EPG) is used to monitor the insect feeding behavior, which can be classified into nonpenetration (np), pathway duration (C), phloem salivation (E1), and phloem ingestion (E2) phases. All EPG recordings are performed for 8 hr. Typical EPG waveforms are displayed in *Figure 2—figure supplement 3*. (**d–f**) Insect performance on transgenic plants overexpressing BtRDP (oeBtRDP). (**d**) Detection of BtRDP in transgenic *Nicotiana tabacum* overexpressing a complete coding region of *BtRDP*. The empty vector (EV) plants are used as the control. The flag tag is fused to the C-terminal ends of recombinant proteins. (**e**) Comparison of insect reproduction on oeBtRDP#1 and EV plants. Five *B. tabaci* individuals are confined to indicated plants for 3 days, and the oviposited eggs are counted. (**f**) Attraction of oeBtRDP#1 and EV leaves to *B. tabaci* in a two-choice equipment. A group of 40 female *B. tabaci* are released into a device containing oeBtRDP#1 and EV leaves. The number of insects settling on each leaf is counted. After 48 hr, the number of eggs on each leaf is counted. (**g–i**) Insect performance on transgenic plants overexpressing BtRDP$^{-sp}$ (oeBtRDP$^{-sp}$). (**g**) Detection of BtRDP$^{-sp}$ in transgenic *N. tabacum* overexpressing BtRDP without a signal peptide. The insect reproduction (**h**) and settlement (**i**) on oeBtRDP$^{-sp}$ transgenic plants are recorded. The EPG data are first checked for normality and homogeneity of variance, and data not fitting a normal distribution are subjected to $\log_{10}$ transformation. Data are presented as mean ± SEM. Statistical significance in panels (**a–c**), (**e**), (**f**), (**h**), and (**i**) was assessed using two-tailed unpaired Student's *t*-tests. ***p < 0.001; **p < 0.01; *p < 0.05; ns, not significant. Western blotting assays are repeated thrice with similar results.

The online version of this article includes the following source data and figure supplement(s) for figure 2:

**Source data 1.** PowerPoint file containing original western blots for *Figure 2a*, indicating the relevant bands and treatments.

**Source data 2.** Original files for western blot analysis displayed in *Figure 2a*.

**Source data 3.** PowerPoint file containing original western blots for *Figure 2d*, indicating the relevant bands and treatments.

**Source data 4.** Original files for western blot analysis displayed in *Figure 2d*.

**Source data 5.** PowerPoint file containing original western blots for *Figure 2g*, indicating the relevant bands and treatments.

**Source data 6.** Original files for western blot analysis displayed in *Figure 2g*.

**Figure supplement 1.** BtRDP is efficiently and specifically suppressed by ds*BtRDP*.

**Figure supplement 1—source data 1.** PowerPoint file containing original western blots for *Figure 2—figure supplement 1b*, indicating the relevant bands and treatments.

**Figure supplement 1—source data 2.** Original files for western blot analysis displayed in *Figure 2—figure supplement 1b*.

*Figure 2 continued on next page*

*Figure 2 continued*

**Figure supplement 2.** Effects of dsRNA treatment on insect survivorship and salivary sheath formation.

**Figure supplement 3.** Typical electrical penetration graph (EPG) waveforms for *Bemisia tabaci* feeding on *Nicotiana tabacum*.

**Figure supplement 4.** Effects of empty vector (EV) transgenic plants on *Bemisia tabaci*.

**Figure supplement 5.** Effects of BtRDP and BtRDP$^{-sp}$ overexpression on *Bemisia tabaci*.

assays further confirmed the specific interaction between BtRDP and NtRLP4 (*Figure 3d*). It was the C-terminal myc tag-fused NtRLP4 (NtRLP4-myc), but not the NtCf9 (NtCf9-myc), that could be immunoprecipitated by the C-terminal flag tag fused BtRDP (BtRDP-flag). In the bimolecular fluorescence complementation (BiFC) assay, YFP fluorescence was observed upon co-expression of the N-terminal nYFP tag fused NtRLP4 and N-terminal cYFP tag fused BtRDP$^{-sp}$ (*Figure 3—figure supplement 3*). Furthermore, in oeBtRDP transgenic plants, endogenous NtRLP4 was specifically immunoprecipitated with BtRDP-flag, which was significantly different from the control (*Figure 3—figure supplement 3*). These results suggest that BtRDP interacts with NtRLP4.

The absence of kinase motifs in the C-terminal region of NtRLP4 suggests that it may function in the cooperation with other proteins to initiate signaling. The receptor-like kinase SOBIR1, which contained a kinase domain (*Figure 3—figure supplement 4*), has been widely reported to be required for the function of LRR-RLPs in the innate immunity (*Gust and Felix, 2014*). However, whether SOBIR1 interacted with malectin-LRR RLP remains largely unknown. Through Co-IP assays, our study demonstrated that NtSOBIR1 was capable of interacting with NtRLP4 but not BtRDP (*Figure 3—figure supplement 4*).

## NtRLP4 and NtSOBIR1 confer resistance to *B. tabaci* infestation

The transgenic *N. tabacum* overexpressing NtRLP4 (oeRLP#1 and oeRLP#2, without any tags) was constructed (*Figure 3e*). Compared with the EV control, both oeRLP lines were less attractive to *B. tabaci*, alongside less settling and ovipositing insects (*Figure 3f*; *Figure 4—figure supplement 1*). Also, *B. tabaci* had a significantly decreased fecundity in oeRLP plants (*Figure 3g*; *Figure 4—figure supplement 1*). Similar results were observed after transient overexpression of NtRLP4-GFP in *N. tabacum* (*Figure 4—figure supplement 1*), indicating that NtRLP4 confers resistance against *B. tabaci*.

Thereafter, the EV and oeRLP#1 plants underwent transcriptomic sequencing (*Figure 4—figure supplement 2*). A total of 729 differentially expressed genes (DEGs) were identified, comprising 423 upregulated and 306 downregulated genes (*Supplementary file 1C*). Enrichment analysis demonstrated that the majority of upregulated DEGs were associated with plant–pathogen interaction, environmental adaptation, MAPK signaling pathway, and signal transduction. In contrast, the glutathione metabolism, carbohydrate metabolism, and amino acid metabolism pathways were significantly enriched in downregulated DEGs (*Figure 4—figure supplement 2*). Noteworthily, numerous DEGs were annotated as RLK/RLP or WRKY transcription factor, with the majority of them being significantly upregulated (*Figure 4—figure supplement 2*). These results suggest an enhanced defense response in oeRLP plants. The crosstalk between salicylic acid (SA) and jasmonic acid (JA) pathways is critical for plant defenses (*Xu et al., 2019*). The increased defense of oeRLP transgenic plants was partially caused by the altered hormonal signaling. The JA-associated gene *FAD7* was significantly induced, while the SA-associated genes *PAL* and *NPR1* were repressed in oeRLP plants (*Figure 3h*). Transient overexpression of NtRLP4-GFP in *N. tabacum* plants via *Agrobacterium* infiltration exerted similar effects (*Figure 4—figure supplement 1*). Furthermore, NtRLP4-GFP significantly enhanced $H_2O_2$ accumulation (*Figure 4—figure supplement 1*).

The function of NtSOBIR1 was also investigated by transient overexpression and hairpin RNAi (*Figure 4—figure supplement 3*). Overexpression of NtSOBIR1-GFP induced the cell death phenotype 5 days post agro-injection (*Figure 4—figure supplement 3*). Insects preferred to settle and oviposit on GFP-expressed *N. tabacum* than on NtSOBIR1-GFP-expressed control, even though no obvious death phenotype was observed 2 days post agro-injection (*Figure 4—figure supplement 3*). For NtSOBIR1 silencing, no cell death phenotype was observed, while the NtSOBIR1-silenced leaves exhibited reduced growth (*Figure 4—figure supplement 3*). Compared with the EV-silenced control,

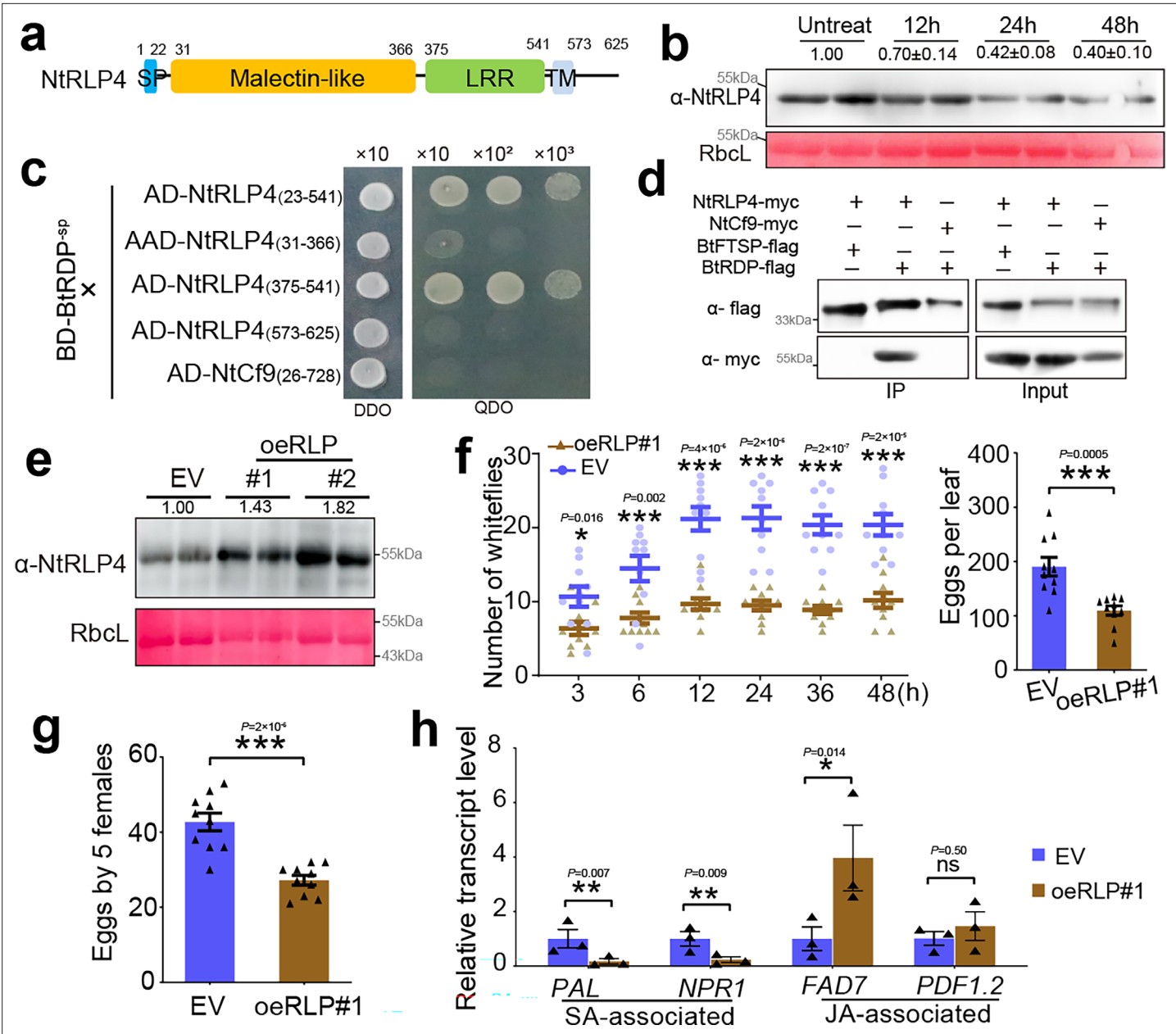

**Figure 3.** NtRLP4 interacts with BtRDP and confers plant resistance to *Bemisia tabaci*. (**a**) Domain organization of NtRLP4. NtRLP4 contains a predicted N-terminal signal peptide (SP), a malectin-like domain, an LRR domain, and a transmembrane domain (TM). (**b**) The protein level of NtRLP4 in response to *B. tabaci* infestation. The untreated *Nicotiana tabacum* is used as a negative control. Four independent biological replicates are performed, and the representative images are displayed. The band density is measured using ImageJ. The density values from four biological replicates are calculated and the mean value in the controls is set at 1.0. (**c**) Yeast two-hybrid assays showing the interaction between BtRDP and NtRLP4. The different combinations of constructs are transformed into yeast cells and are grown on the selective medium SD/-Trp/-Leu (DDO), and the interactions are tested with SD/-Trp/-Leu/-His/-Ade (QDO). (**d**) Co-immunoprecipitation assay showing the interaction between BtRDP and NtRLP4. Total proteins are extracted from *N. benthamiana* leaves transiently co-expressing NtRLP4-myc/NtCf9-myc with BtFTSP-flag/BtRDP-flag. All genes are expressed with a complete coding region, and the myc/flag tags are fused at the C-terminus. Precipitation is performed using flag beads. The samples are probed with anti-flag and anti-myc antibodies for immunoblotting analysis. (**e–h**) Analysis of transgenic *N. tabacum* overexpressing NtRLP4 (oeRLP). (**e**) Detection of NtRLP4 level in oeRLP plants. The empty vector (EV) plant is used as a control. Two independent oeRLP lines are selected. The samples are probed with an anti-NtRLP4 antibody. Rubisco staining (RbcL) is used to visualize the amount of sample loading. (**f**) Attraction of oeRLP#1 and EV leaves to *B. tabaci* in a two-choice equipment. A group of 40 female *B. tabaci* are released into a device containing oeRLP#1 and EV leaves. The number of insects settling on each leaf is counted at each time point. After 48 hr, the number of eggs on each leaf is counted. (**g**) Comparison of insect reproduction on oeRLP#1 and EV plants. Five *B. tabaci* individuals are confined to indicated plants for 3 days, and the oviposited eggs are counted. (**h**) Relative transcript levels of salicylic acid (SA)- and jasmonic acid (JA)-associated genes in oeRLP#1 and EV plants. PAL, phenylalanine ammonia lyase; NPR1, nonexpressor of pathogenesis-

*Figure 3 continued on next page*

*Figure 3 continued*

related protein 1; FAD7, fatty acid desaturase 7; PDF1.2, plant defensin 1.2. Two independent biological replicates are performed in (**d**) and (**e**). Data in (**f**–**h**) are presented as mean values ± SEM. For insect bioassays in (**f**) and (**g**), $n = 10$ independent biological replicates. For quantitative real-time PCR (qRT-PCR) in (**h**), $n = 3$ independent biological replicates. p-values are determined by two-tailed unpaired Student's *t*-test. ***$p < 0.001$; **$p < 0.01$; *$p < 0.05$; ns, not significant.

The online version of this article includes the following source data and figure supplement(s) for figure 3:

**Source data 1.** PowerPoint file containing original western blots for *Figure 3b*, indicating the relevant bands and treatments.

**Source data 2.** Original files for western blot analysis displayed in *Figure 3b*.

**Source data 3.** PowerPoint file containing original western blots for *Figure 3d*, indicating the relevant bands and treatments.

**Source data 4.** Original files for western blot analysis displayed in *Figure 3d*.

**Source data 5.** PowerPoint file containing original western blots for *Figure 3e*, indicating the relevant bands and treatments.

**Source data 6.** Original files for western blot analysis displayed in *Figure 3e*.

**Figure supplement 1.** Expression patterns of *NtRLP4* in response to *Bemisia tabaci* infestation.

**Figure supplement 2.** Sequence alignments of SlCf9 and NtCf9.

**Figure supplement 3.** BtRDP interacts with NtRLP4.

**Figure supplement 3—source data 1.** PowerPoint file containing original western blots for *Figure 3—figure supplement 3b*, indicating the relevant bands and treatments.

**Figure supplement 3—source data 2.** Original files for western blot analysis displayed in *Figure 3—figure supplement 3b*.

**Figure supplement 3—source data 3.** PowerPoint file containing original western blots for *Figure 3—figure supplement 3d*, indicating the relevant bands and treatments.

**Figure supplement 3—source data 4.** Original files for western blot analysis displayed in *Figure 3—figure supplement 3d*.

**Figure supplement 4.** NtSOBIR1 interacts with NtRLP4 but not BtRDP.

**Figure supplement 4—source data 1.** PowerPoint file containing original western blots for *Figure 3—figure supplement 4b*, indicating the relevant bands and treatments.

**Figure supplement 4—source data 2.** Original files for western blot analysis displayed in *Figure 3—figure supplement 4b*.

**Figure supplement 4—source data 3.** PowerPoint file containing original western blots for *Figure 3—figure supplement 4d*, indicating the relevant bands and treatments.

**Figure supplement 4—source data 4.** Original files for western blot analysis displayed in *Figure 3—figure supplement 4d*.

---

NtSOBIR1-silenced *N. tabacum* was slightly attractive to *B. tabaci* (*Figure 4—figure supplement 3*). Collectively, these results demonstrate that NtSOBIR1 confers resistance to *B. tabaci* infestation.

## BtRDP attenuates plant defenses by promoting the degradation of host RLP4

To investigate the interplay between NtRLP4 and BtRDP, transgenic *N. tabacum* silencing NtRLP4 (*RNAi-RLP*#1 and *RNAi-RLP*#2) plants were constructed. Both *RNAi-RLP* lines showed reduced NtRLP4 levels compared with EV plants, with *RNAi-RLP*#2 exhibiting a stronger silencing effect (*Figure 5—figure supplement 1*). Compared with EV plants, *B. tabaci* showed a tendency to settle on NtRLP4-silenced plants and lay more eggs (*Figure 4—figure supplement 4*). Thereafter, ds*GFP*- and ds*BtRDP*-treated *B. tabaci* were reared on EV, *RNAi-RLP*#1, and *RNAi-RLP*#2, respectively. The ds*BtRDP*-treated *B. tabaci* produced less offspring than the control (decreased by 28.3%; p = 0.0007) on EV plants (*Figure 4a*), consistent with the results obtained from WT plants (*Figure 2b*). On the *RNAi-RLP*#1 plant, the impaired insect reproduction was still observed after ds*BtRDP* treatment, but the extent was slightly reduced (decreased by 23.7%; p = 0.002) (*Figure 4a*). On the *RNAi-RLP*#2 plant, there was no significant difference between ds*GFP* and ds*BtRDP* treatments (*Figure 4a*). The differential rescue effect between the two RNAi lines might result from their different NtRLP4 silencing efficiencies, with the lower NtRLP4 level in *RNAi-RLP*#2 leading to the more complete rescue pheno-type (*Figure 4a*). Collectively, these results suggest that *RNAi-RLP* plants are beneficial for insects and can partially, but not completely, rescue the impaired feeding performance of ds*BtRDP*-treated insects.

To investigate the influence of BtRDP on NtRLP4, GFP-tagged NtRLP4 (NtRLP4-GFP) was tran-siently co-expressed with mCherry-tagged BtRDP (BtRDP-mCherry), red fluorescent protein

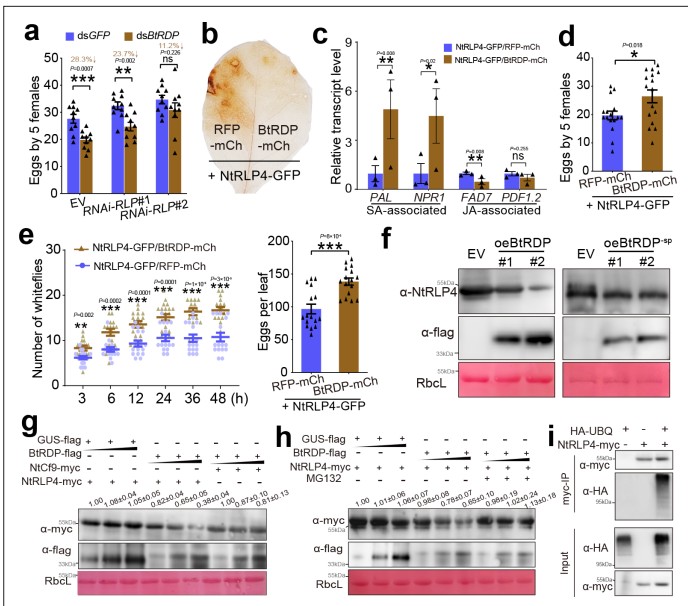

**Figure 4.** BtRDP suppresses plant defenses by promoting NtRLP4 degradation. (**a**) Effects of ds*BtRDP* suppression on insect reproduction (*n* = 10 independent biological replicates) when feeding on empty vector (EV) and NtRLP4-silenced (*RNAi-RLP*) transgenic *Nicotiana tabacum*. (**b**) Transient overexpressing BtRDP-mCherry attenuates $H_2O_2$ accumulation caused by NtRLP4-GFP overexpression. *N. tabacum* co-expressing RFP-mCherry and NtRLP4-GFP is used as a control. All genes are expressed with a complete coding region. The experiment is repeated five times with similar results. (**c**) Relative transcript level of salicylic acid (SA)- and jasmonic acid (JA)-associated genes in BtRDP-mCherry/NtRLP4-GFP and RFP-mCherry/NtRLP4-GFP plants (*n* = 3 independent biological replicates). PAL, phenylalanine ammonia lyase; NPR1, nonexpressor of pathogenesis-related protein 1; FAD7, fatty acid desaturase 7; PDF1.2, plant defensin 1.2. (**d**) Comparison of insect reproduction (*n* = 16 independent biological replicates) on transient-expressed BtRDP-mCherry/NtRLP4-GFP and RFP-mCherry/NtRLP4-GFP plants. (**e**) Attraction of BtRDP-mCherry/NtRLP4-GFP and RFP-mCherry/NtRLP4-GFP leaves to *B. tabaci* in a two-choice equipment (*n* = 16 independent biological replicates). The number of eggs on each leaf is counted at 48 hr post insect release. (**f**) Effects of BtRDP and BtRDP^-sp^ overexpression on NtRLP4 accumulation. Transgenic oeBtRDP (left) and oeBtRDP^-sp^ (right) plants are probed with anti-NtRLP4 and anti-flag antibodies for immunoblotting analysis. The experiments are repeated twice with similar results. (**g**) Degradation of NtRLP4 by BtRDP in *N. benthamiana* leaves. NtRLP4-myc and BtRDP-flag are transiently co-expressed in *N. benthamiana* plants through *Agrobacterium* infiltration. *Agrobacterium* carrying NtCf9-myc and GUS-flag is used as negative controls. (**h**) Effects of 26S proteasome inhibitor (MG132) on NtRLP4 accumulation. Co-infiltrated leaves are treated with MG132 at 24 hr post-injection. The samples are probed with anti-flag and anti-myc antibodies for immunoblot analysis. Rubisco staining (RbcL) is conducted to visualize the amount of sample loading. The small triangle indicates the different concentrations of *Agrobacterium* (OD₆₀₀ = 0.05, 0.3, and 1.0). (**i**) NtRLP4 is ubiquitinated in planta. NtRLP4-myc is co-expressed transiently with HA-UBQ in *N. benthamiana* leaves. Extracted total proteins are immunoprecipitated by anti-myc beads and immunoblotted with anti-myc or anti-HA antibody. Experiments are repeated thrice for (**g**) and (**h**), while twice for (**i**). Band density is measured using ImageJ. The density values from three biological replicates are calculated and the mean value in the first lane is set at 1.0. Data are presented as mean values ± SEM. p-values are determined by two-tailed unpaired Student's *t*-test. ***p < 0.001; **p < 0.01; *p < 0.05; ns, not significant.

The online version of this article includes the following source data and figure supplement(s) for figure 4:

**Source data 1.** PowerPoint file containing original western blots for *Figure 4f*, indicating the relevant bands and treatments.

**Source data 2.** Original files for western blot analysis displayed in *Figure 4f*.

**Source data 3.** PowerPoint file containing original western blots for *Figure 4g*, indicating the relevant bands and treatments.

**Source data 4.** Original files for western blot analysis displayed in *Figure 4g*.

**Source data 5.** PowerPoint file containing original western blots for *Figure 4h*, indicating the relevant bands and treatments.

**Source data 6.** Original files for western blot analysis displayed in *Figure 4h*.

*Figure 4 continued on next page*

*Figure 4 continued*

**Source data 7.** PowerPoint file containing original western blots for *Figure 4i*, indicating the relevant bands and treatments.

**Source data 8.** Original files for western blot analysis displayed in *Figure 4i*.

**Figure supplement 1.** Effect of NtRLP4 overexpression on *Bemisia tabaci* performance.

**Figure supplement 2.** Transcriptomic comparison of empty vector (EV) and oeNtRLP4#1 (oeRLP#1) transgenic plants.

**Figure supplement 3.** Effect of NtSOBIR1 overexpression and silencing on *Bemisia tabaci* performance.

**Figure supplement 4.** Effects of *RLP* silencing on *Bemisia tabaci* performance.

**Figure supplement 4—source data 1.** PowerPoint file containing original western blots for *Figure 4—figure supplement 4a*, indicating the relevant bands and treatments.

**Figure supplement 4—source data 2.** Original files for western blot analysis displayed in *Figure 4—figure supplement 4a*.

**Figure supplement 5.** Influence of BtRDP on NtRLP4 accumulation by fluorescent analysis.

**Figure supplement 6.** Effects of BtRDP on suppressing NtRLP4-associated plant defenses.

**Figure supplement 7.** Degradation of NtRLP4 by purified BtRDP$^{-sp}$ in *Nicotiana benthamiana* leaves.

**Figure supplement 7—source data 1.** PowerPoint file containing original western blots for *Figure 4—figure supplement 7*, indicating the relevant bands and treatments.

**Figure supplement 7—source data 2.** Original files for western blot analysis displayed in *Figure 4—figure supplement 7*.

**Figure supplement 8.** Effect of BtRDP on NtRLP4 and NtSOBIR1.

**Figure supplement 8—source data 1.** PowerPoint file containing original western blots for *Figure 4—figure supplement 8*, indicating the relevant bands and treatments.

**Figure supplement 8—source data 2.** Original files for western blot analysis displayed in *Figure 4—figure supplement 8*.

**Figure supplement 9.** Effect of BtRDP on the transcript level of *NtRLP4*.

**Figure supplement 10.** Effects of autophagy inhibitor on NtRLP4 accumulation.

**Figure supplement 10—source data 1.** PowerPoint file containing original western blots for *Figure 4—figure supplement 10*, indicating the relevant bands and treatments.

**Figure supplement 10—source data 2.** Original files for western blot analysis displayed in *Figure 4—figure supplement 10*.

(RFP)-mCherry, and BtFTSP1-mCherry through agroinfiltration, respectively. All tags were fused with the C-terminal region. As discovered, BtRDP-mCherry did not influence the location patterns of NtRLP4-GFP (*Figure 4—figure supplement 5*). However, a significant reduction in the fluorescent signal of NtRLP4-GFP was detected after co-expression with BtRDP-mCherry, but not with the controls (*Figure 4—figure supplement 5*). Besides, co-expression with BtRDP-mCherry/NtRLP4-GFP significantly reduced $H_2O_2$ accumulation (*Figure 4b*), suppressed the expression of JA-associated genes *FAD7* and *PDF.12*, and induced that of the SA-associated genes *PAL* and *NPR1* (*Figure 4c*). It was discovered by insect bioassays that transient overexpression of BtRDP-mCherry rescued the negative effects on whitefly performance caused by NtRLP4-GFP overexpression (*Figure 4d, e*). In the host choice experiments, *B. tabaci* displayed a preference for settling on *N. tabacum* leaves co-expressing BtRDP-mCherry/NtRLP4-GFP over those co-expressing RFP-mCherry/NtRLP4-GFP (*Figure 4d*). However, almost no preference was observed in *B. tabaci* feeding on *N. tabacum* plants co-expressing BtRDP-mCherry/NtRLP4-GFP and plants expressing GFP alone (*Figure 4—figure supplement 6*).

The attenuated NtRLP4-GFP fluorescent signal (*Figure 4—figure supplement 5*), along with the observed decrease in the NtRLP4 protein level during *B. tabaci* infestation (*Figure 3b*), led us to speculate on whether BtRDP influenced the accumulation of NtRLP4. To this end, the BtRDP$^{-sp}$-his and GFP-his proteins were prokaryotically expressed in *Escherichia coli*. Different concentrations of purified BtRDP$^{-sp}$-his and GFP-his proteins were infiltrated into *N. tabacum* plants overexpressing NtRLP4-myc. Interestingly, the protein level of NtRLP4-myc slightly decreased with the increasing amount of BtRDP$^{-sp}$-his, while no such effect was observed with GFP-his (*Figure 4—figure supplement 7*). Moreover, the

BtRDP⁻ˢᵖ-his protein did not have any significant effect on the accumulation of another RLP NtCf9-like (*Figure 4—figure supplement 8*). We further assessed the impact of BtRDP on NtSOBIR1 following NtRLP4 destabilization. As the BtRDP-flag amount increased, NtRLP4-myc accumulation was markedly reduced, while NtSOBIR1-flag levels remained unchanged (*Figure 4—figure supplement 9*).

Then, the NtRLP4 levels in oeBtRDP and oeBtRDP⁻ˢᵖ plants were detected. Compared with the EV control, a reduced NtRLP4 abundance was observed in both oeBtRDP#1 and oeBtRDP#2 transgenic lines, but not in the oeBtRDP⁻ˢᵖ plants (*Figure 4f*), suggesting the importance of BtRDP secretion in extracellular space. BtRDP attached to the salivary sheath localized in plant apoplast (*Figure 1f*). The presence of signal peptide promoted the secretion of BtRDP into the extracellular space. Therefore, the complete coding region of *BtRDP* was used in subsequent experiments. The NtRLP4-myc and BtRDP-flag were transiently co-expressed in *N. benthamiana* plants through *Agrobacterium* infiltration. As a result, BtRDP-flag significantly decreased the protein level of NtRLP4-myc, but not NtCf9-myc (*Figure 4g*). qRT-PCR analyses demonstrated that the transcript level of *NtRLP4* remained unchanged (*Figure 4—figure supplement 9*). These findings indicate that the reduction in the protein level of NtRLP4 is not caused by transcriptional changes but rather by degradation.

The ubiquitin system and autophagy are two major pathways regulating protein degradation (*Varshavsky, 2017*). At first, MG132, a 26S proteasome inhibitor known to inhibit the degradation of ubiquitin-conjugated proteins, was employed (*Li et al., 2022*). The results showed that MG132 efficiently prevented the degradation of NtRLP4-myc by BtRDP-flag (*Figure 4h*). Then, the autophagy inhibitor 3-Methyladenine (3-MA) and bafilomycin A1 (BAF) that can block autophagy in plants were used (*Takatsuka et al., 2004*). However, the decreased protein level of NtRLP4-myc was still observed in the presence of 3-MA or BAF (*Figure 4—figure supplement 10*). Additionally, NtRLP4-myc and HA-tagged ubiquitin (HA-UBQ) were co-expressed in *N. benthamiana*. Total proteins were extracted from *N. benthamiana* leaves and then subjected to immunoprecipitation with anti-myc beads. The ubiquitin attached to NtRLP4 was determined by immunoblot using the anti-HA antibody. Leaves expressing NtRLP4-myc or UBQ-HA along were used as negative controls. The results showed that a strong UBQ signal was detected in the immunoprecipitated product only when NtRLP4-myc and UBQ-HA were co-expressed (*Figure 4i*). The molecular weight of UBQ-attached NtRLP4-myc was remarkably larger than that of NtRLP4-myc itself (*Figure 4i*). These results suggest that BtRDP promotes NtRLP4 degradation through the ubiquitin system, but not the autophagy pathway.

## Degrading plant RLP4s by salivary effectors is a common strategy for insects

To investigate the ability of BtRDP to degrade RLPs from other host plants of *B. tabaci*, *Solanum lycopersicum* RLP4 (SlRLP4; GenBank accession: XP_004232910) was selected for further analysis. Sequence analysis revealed a similar gene structure between SlRLP4 and NtRLP4 (*Figure 5—figure supplement 1*; *Figure 5—figure supplement 2*). Moreover, the interaction between BtRDP and SlRLP4 was confirmed through Y2H and Co-IP assays (*Figure 5—figure supplement 2*). Additionally, SlRLP4-myc and SlCf9-like-myc were transiently co-expressed with varying concentrations of BtRDP-flag or GUS-flag, with the myc and flag tags fused with the C-terminus. The results showed that BtRDP-flag exerted no influence on the accumulation of SlCf9-like-myc protein. In contrast, the protein level of SlRLP4-myc was significantly reduced in the presence of BtRDP-flag (*Figure 5—figure supplement 2*), indicating the conserved function of BtRDP in *B. tabaci* adaptation to different host plants.

BtRDP was a gene specific to the Aleyrodidae family, lacking homologs in other species. To explore whether the degradation of RLPs by salivary effectors was a conserved mechanism among different herbivorous insects, an RLP4 homology from *Oryza sativa* (OsRLP4; GenBank accession: XP_015645303; *Figure 5a*) and 20 salivary proteins from planthopper species lacking signal peptides were paired to identify the potential interactions using Y2H assays. As a result, only the salivary protein NlSP104⁻ˢᵖ (*N. lugens* salivary protein consisting of 104 amino acids, GenBank accession: MF278694) interacted with OsRLP4₍₂₉₋₅₅₁₎ (OsRLP4 lacking signal peptides and transmembrane domains), while the other salivary proteins tested did not show any interaction (*Figure 5b, c*; *Figure 5—figure supplement 3*). NlSP104 did not exhibit any sequence or structural similarity to BtRDP (*Figure 5—figure supplement 4*). Truncation analysis revealed that NlSP104⁻ˢᵖ interacted with both the LRR domain and the malectin-like domain of OsRLP4 (*Figure 5c*), which was slightly different from the BtRDP-NtRLP4 interaction (*Figure 3c*). Given that both BtRDP and NlSP104 interacted with the LRR domain of RLP4, further

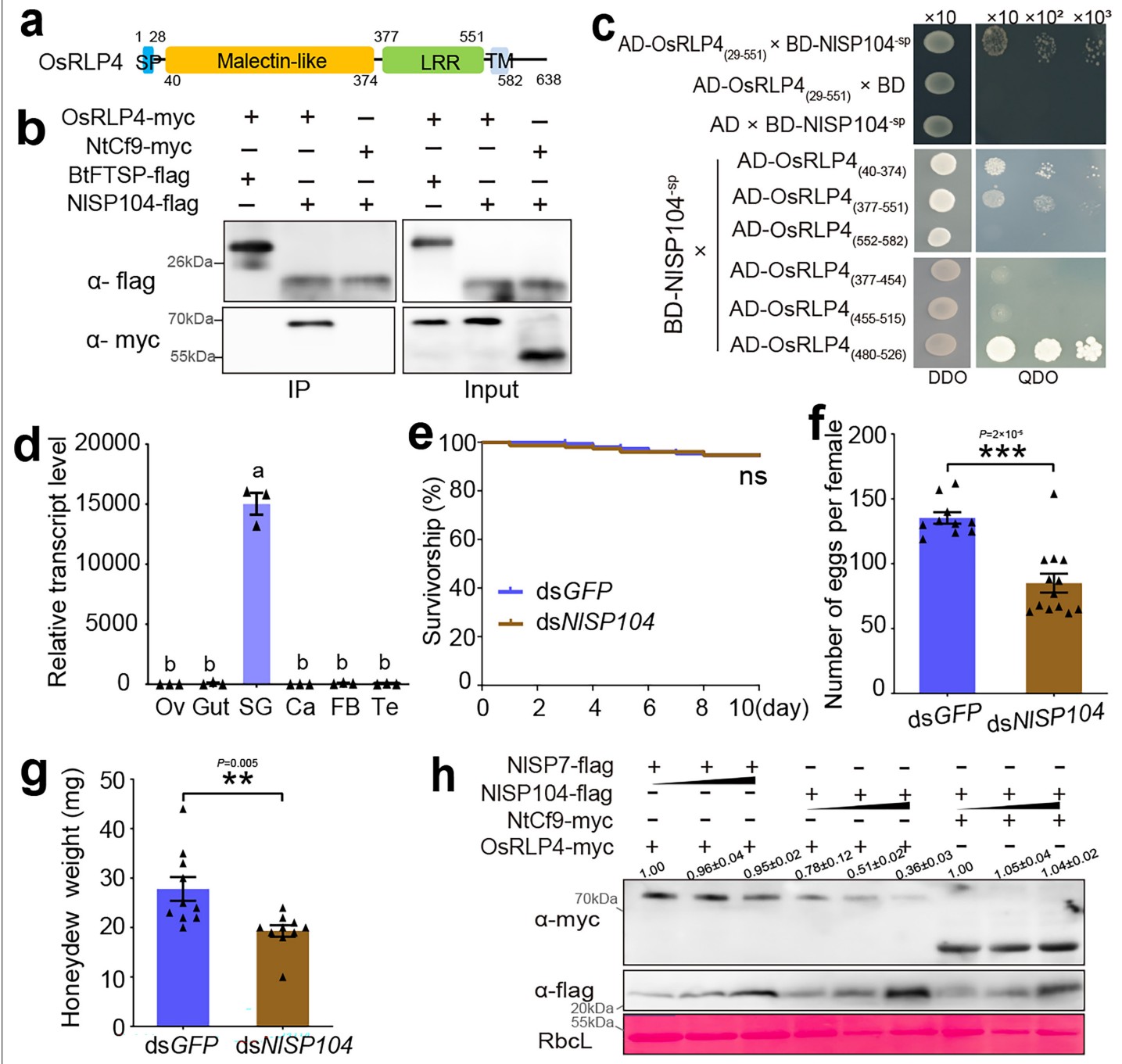

**Figure 5.** Rice RLP4 is targeted by salivary protein NISP104 from *Nilaparvata lugens*. (**a**) Domain organization of *Oryza sativa* RLP4 (OsRLP4). (**b, c**) Yeast two-hybrid and Co-IP assays showing the interaction between OsRLP4 and NISP104. All genes are expressed with a complete coding region in Co-IP assays, while NISP104 without a signal peptide is used in yeast two-hybrid assays. Experiments in (**b**) are repeated twice with the similar result. (**d**) The expression patterns of *NISP104* in different *N. lugens* tissues. Te, testis; Ov, ovary; SG, salivary gland; Ca, carcass; FB, fat body. Data are presented as mean values ± SEM (n = 3 independent biological replicates). Different lowercase letters indicate statistically significant differences at p < 0.05 level according to one-way ANOVA test followed by Tukey's multiple comparisons test. Effects of dsRNA treatment on insect survivorship (**e**), reproduction (**f**), and honeydew excretion (**g**). For survivorship analysis, a group of 30 *N. lugens* are reared in a cage. Three independent biological replications are performed. Differences in survivorship between the two treatments are tested by log-rank test. ns, not significant. For reproduction analysis, n = 10 and 13 individuals are tested in ds*GFP*- and ds*NISP104*-treatment, respectively. The sterile females are excluded from data analysis. For honeydew analysis, n = 10 independent biological replicates. p-values in (**f**) and (**g**) are determined by two-tailed unpaired Student's *t*-test. ***p < 0.001; **p < 0.01. (**h**) Effect of NISP104 on the accumulation of transient-expressed OsRLP4. OsRLP4-myc is agro-injected into *Nicotiana benthamiana* together with different concentrations of NISP104-flag and NISP7-flag. NtCf9-myc is used as a negative control. The small triangle indicates the different concentrations ($OD_{600}$

*Figure 5 continued on next page*

*Figure 5 continued*

= 0.05, 0.3, and 1.0) of *Agrobacterium*. Rubisco staining (RbcL) is conducted to visualize the amount of sample loading. Experiments are repeated three times with similar results. Band density is measured using ImageJ. The density values from three biological replicates are calculated with the mean value in the first lane being set at 1.0.

The online version of this article includes the following source data and figure supplement(s) for figure 5:

**Source data 1.** PowerPoint file containing original western blots for *Figure 5b*, indicating the relevant bands and treatments.

**Source data 2.** Original files for western blot analysis displayed in *Figure 5b*.

**Source data 3.** PowerPoint file containing original western blots for *Figure 5h*, indicating the relevant bands and treatments.

**Source data 4.** Original files for western blot analysis displayed in *Figure 5h*.

**Figure supplement 1.** Sequence alignment and phylogenetic tree of RLP4 homologs.

**Figure supplement 2.** Influence of BtRDP on RLP4 homology in *Solanum lycopersicum*.

**Figure supplement 2—source data 1.** PowerPoint file containing original western blots for *Figure 4—figure supplement 7c*, indicating the relevant bands and treatments.

**Figure supplement 2—source data 2.** Original files for western blot analysis displayed in *Figure 4—figure supplement 7c*.

**Figure supplement 2—source data 3.** PowerPoint file containing original western blots for *Figure 4—figure supplement 7d*, indicating the relevant bands and treatments.

**Figure supplement 2—source data 4.** Original files for western blot analysis displayed in *Figure 4—figure supplement 7d*.

**Figure supplement 3.** Yeast two-hybrid assays showing the interaction between OsRLP4 and salivary proteins from planthopper species.

**Figure supplement 4.** Three-dimensional structure of BtRDP and NlSP104.

**Figure supplement 5.** Expression patterns of *Nilaparvata lugens* NlSP104.

investigation focused on the specific region of the LRR domain responsible for NlSP104 binding. The LRR domain of OsRLP4 was divided into three segments, mirroring the division of the LRR domain in NtRLP4 (*Figure 3—figure supplement 3*). NlSP104$^{-sp}$ interacted with the truncated LRR domain (OsRLP4$_{(480–526)}$) (*Figure 5c*). In contrast, none of the truncated LRR domain of NtRLP4 interacted with BtRDP$^{-sp}$ (*Figure 3—figure supplement 3*). Collectively, these findings suggest that the binding sites for salivary proteins to RLP4 may differ between *N. lugens* and *B. tabaci*.

NlSP104 was a planthopper-specific salivary protein nearly exclusively expressed in salivary glands (*Supplementary file 1A*; *Figure 5d*; *Figure 5—figure supplement 5*). The function of NlSP104 or its gene homologs in other planthopper species has not been well investigated in previous studies. Silencing NlSP104 did not affect insect survivorship (*Figure 5e*). However, significant reductions in honeydew excretion and impaired fecundity were observed in ds*NlSP104*-treated *N. lugens* (*Figure 5f, g*), indicating the importance of NlSP104 in insect feeding.

Subsequently, OsRLP4-myc was transiently co-expressed with a complete coding region of *NlSP104* (C-terminal flag tagged, NlSP104-flag). Another salivary protein NlSP7-flag and NtCf9-myc were used as negative controls. It was found that NlSP104-flag specifically and significantly decreased the protein level of OsRLP4-myc, which differed significantly from that of the NlSP7-flag (*Figure 5h*). Additionally, NlSP104-flag failed to decrease the protein level of NtCf9-myc (*Figure 5h*). These findings suggest that *N. lugens* potentially evolves salivary effectors to target rice RLPs.

## Discussion

Plants have evolved a complex innate immune system to protect themselves against pathogen/insect invasion, and the cell surface-localized PRRs function on the frontline (*Macho and Zipfel, 2014*). To counteract this response, the pathogens have developed various strategies to avoid plant PRR-mediated PTI for their own advantage (*Tariqjaveed et al., 2021*). However, it remains unknown whether insects take the similar strategy to sabotage plant immunity. In this study, we demonstrate that the RLP4 confers plant resistance against herbivorous insects. In response, both planthoppers and whiteflies secrete salivary effectors to facilitate the degradation of defensive RLP4, leading to the attenuated plant immunity. Our results suggest that manipulation of PRRs may be a conserved strategy for herbivorous insects.

PRRs detect modified-self or non-self patterns and initiate a cascade of immune signaling events, which subsequently regulate downstream defense responses, conferring plant resistance against

invaders (*Macho and Zipfel, 2014*; *Zipfel, 2014*). Although various insect salivary components have been identified as triggers for plant immune responses, the specific PRRs involved in this process remain unclear. Our study indicates that RLP4 significantly influences insect performance by modulating plant hormones and ROS (*Figure 3f–h*; *Figure 4—figure supplement 1*), highlighting the critical role of RLP4 in plant immunity. However, the precise ligand for RLP4 has yet to be identified. The malectin-LRR receptor RLP4 may respond to the DAMPs or HAMPs released during insect feeding (*Doblas et al., 2018*), a hypothesis that warrants further investigation. RLP4 lacks an intracellular kinase domain and instead forms bimolecular receptor kinases by interacting with SOBIR1 (*Figure 3—figure supplement 4*). Silencing of NtRLP4 or NtSOBIR1 in *N. tabacum* attracts *B. tabaci* and promotes insect reproduction, whereas overexpression of either gene exerts the opposite effect (*Figure 4*). The association between RLPs and SOBIR1 protein kinases is structurally and functionally analogous to that of genuine receptor kinases (*Gust and Felix, 2014*). Therefore, destabilizing the RLP4/SOBIR1 complex can potentially disrupt the immune recognition processes. Noteworthily, although NtRLP4 interacts with SOBIR1, this alone does not confirm that it operates strictly through this canonical module. Evidence from other RLPs shows that co-receptor usage can be flexible, and some RLPs function partly or conditionally independent of SOBIR1 (*Albert et al., 2015*). Therefore, a more definitive assessment of NtRLP4 signaling will therefore require genetic dissection of its co-receptor dependencies, including but not limited to SOBIR1.

As pivotal regulators in immune signaling, PRRs are frequently targeted by multiple pathogens through diverse mechanisms. For example, *Pseudomonas syringae* hampers the functionality of receptor complexes by disrupting the ligand-dependent association between receptor-like kinase BAK1 and the flagellin receptor FLS2 (*Shan et al., 2008*), while *Phytophthora infestans* promotes alternative splicing of the host RLPKs to enhance infection (*Huang et al., 2020*). PRRs are usually subjected to degradation via post-translational modifications (*Li et al., 2014*). In plants, the activity and stability of PRRs are under stringent regulation since excessive activation of PRRs could be detrimental (*Zhang and Zeng, 2020*; *Chen et al., 2022*). One prevalent regulatory mechanism involves ubiquitination followed by subsequent degradation mediated by the 26S proteasome (*Li et al., 2014*). Mounting evidence has demonstrated that bacteria, fungi, viruses, and oomycetes exploit such mechanisms to facilitate PRR degradation (*Langin et al., 2023*; *Rosas-Diaz et al., 2018*; *Irieda et al., 2019*; *Qin et al., 2018*). Nevertheless, the occurrence of effector-mediated PRR degradation in insects is unreported. Our study reveals that the salivary effectors BtRDP from *B. tabaci* and NlSP104 from *N. lugens* interact with plant RLP4s, resulting in their degradation in a ubiquitin-dependent manner (*Figure 4f–i*).

Notably, BtRDP and NlSP104 exhibit neither sequence nor structural similarity and do not resemble the known eukaryotic ubiquitin-ligase domains. Their interaction with RLP4s occurs in the extracellular space (*Figures 3d and 5c*), whereas the ubiquitin–proteasome system primarily functions within the cytosol and nucleus (*Xu and Xue, 2019*). Furthermore, NtRLP4 reduction is observed only in oeBtRDP transgenic plants, not in oeBtRDP⁻ˢᵖ plants (*Figure 4f*), indicating that BtRDP exerts its influence on NtRLP4 in the extracellular space. These observations collectively refute the possibility that BtRDP or NlSP104 possesses the intrinsic E3 ligase activity capable of directly ubiquitinating RLP4s within plant cells. Importantly, the reduced NtRLP4 levels may not stem from a direct physical interaction between BtRDP and NtRLP4. Instead, BtRDP may indirectly affect RLP4 post-translational modification, thereby accelerating its degradation, which warrants further investigation. Additionally, the independent evolution of RLP4-targeting effectors in various insect lineages may expedite plant-insect co-evolution. Plants with robust RLP4-mediated defenses can select for herbivores carrying effective effectors, driving an evolutionary 'arms race'. Conversely, insects targeting similar host proteins may exert parallel selective pressures on unrelated plant species, potentially shaping defense diversification across plant communities. Understanding these dynamics provides insight into how molecular interactions scale up to influence herbivore adaptation, host range, and the ecological outcomes of plant–insect interactions.

Herbivorous insects employ a mix of salivary components to ensure successful feeding (*Huang et al., 2023*). Previous research has indicated that *B. tabaci* suppresses plant defenses by utilizing SA–JA crosstalk, with the salivary protein Bt56 playing a crucial role in this process (*Xu et al., 2019*). Bt56 activates the SA-signaling pathway by directly interacting with a KNOTTED 1-like homeobox transcription factor (*Xu et al., 2019*). This study investigates the effects of another salivary effector,

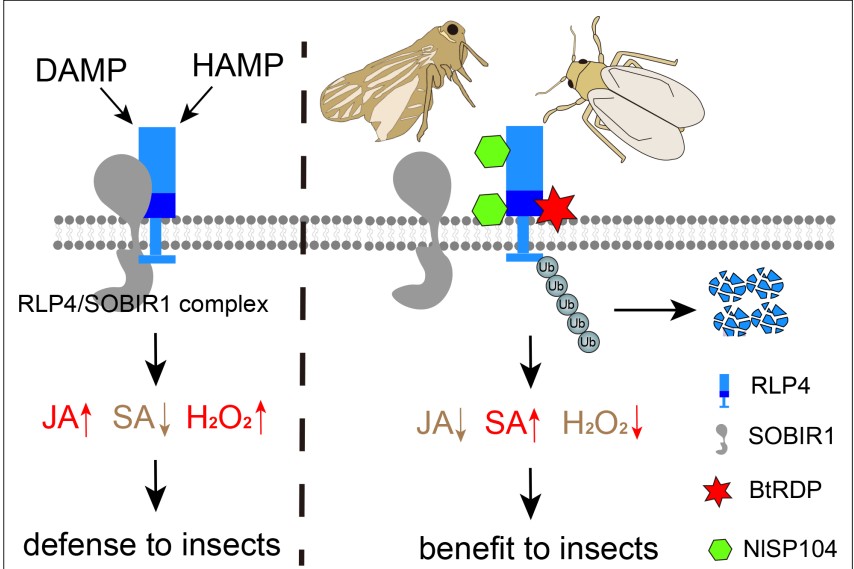

**Figure 6.** The proposed model for the suppression of receptor-like protein (RLP)-mediated plant defenses by salivary effectors. Host plants employ pattern-recognition receptors (PRRs) to detect various damage-associated molecular patterns (DAMPs) and herbivore-associated molecular patterns (HAMPs) triggered by insect feeding. The RLP4/SOBIR1 complex plays a vital role in initiating pattern-triggered immunity (PTI), including $H_2O_2$ burst, upregulation of jasmonic acid (JA), and downregulation of salicylic acid (SA), which hinders insect feeding. The whitefly *Bemisia tabaci* and planthopper *Nilaparvata lugens* independently evolved salivary proteins that targeted plant RLP4. *B. tabaci* salivary sheath protein BtRDP interacts with the leucine-rich repeat (LRR) domain of RLP4 from *Nicotiana tabaci* and *Solanum lycopersicum*, while *N. lugens* NlSP104 targets both the LRR domain and the malectin-like domain of *Oryza sativa* RLP4. These interactions promote the ubiquitin-dependent degradation of RLP4, thereby disrupting the stability of the RLP4/SOBIR1 complex. The presence of salivary effectors causes a hormonal shift and suppresses the $H_2O_2$ burst, finally favoring insect feeding.

BtRDP, on plant hormones and finds that it elicits similar effects to Bt56, albeit primarily impacting the signal recognition process (*Figure 4*). This leads us to speculate that different salivary effectors from the same insect species may simultaneously target the same signaling pathway to ensure effective immune suppression. Our analysis reveals that BtRDP interacts with other host genes apart from NtRLP4 (*Supplementary file 1B*), and silencing NtRLP4 cannot completely rescue the impaired feeding performance of ds*BtRDP*-treated insects (*Figure 4a*). Salivary proteins exert a crucial effect on mediating the interactions between herbivores and plants. The dynamic interplay between plants and insects may be highly complex; one salivary protein can target multiple plant pathways, and one plant pathway is influenced by multiple salivary proteins.

Together, this study reveals that suppressing PRR-mediated plant immunity may be a conserved strategy employed by herbivorous insects for successful feeding. We demonstrate that whiteflies and planthoppers have independently evolved salivary effectors that facilitate the ubiquitin-dependent degradation of defensive RLP4 in host plants, thereby dampening RLP4-mediated plant immunity (*Figure 6*). Nevertheless, the precise mechanism by which RLP4 contributes to plant defense warrants further consideration. While it may function as a canonical PRR that perceives insect-derived molecular patterns, several lines of evidence point to an alternative interpretation. Structurally, RLP4 differs from classical LRR-RLP: it contains a malectin-like domain and a relatively small LRR domain, contrasting with typical LRR-RLPs that often possess large LRRs dedicated to ligand binding. Functionally, NtRLP4 overexpression lines exhibit significantly altered transcriptional profiles and dysregulated SA/JA pathways even in the absence of insect infestation, a phenotype inconsistent with canonical PRRs, which typically remain quiescent until ligand perception. These findings point to an alternative explanation: rather than functioning as a classical PRR that recognizes insect-derived molecules, RLP4 may act as a regulatory component within plant immune signaling networks. Elucidating the precise mechanism of RLP4 in conferring plant defense against herbivorous insects will therefore be an important focus of future research.

## Materials and methods

### Insects and plants

A colony of *B. tabaci* (cryptic species MED) was originally collected from a soybean field in Suzhou, Anhui Province, China. The insects were subsequently maintained on *N. tabacum* cv. K326 plants under controlled laboratory conditions (25 ± 1°C, 50–70% relative humidity, 16:8 hr light:dark photoperiod). For experimental use, *N. benthamiana* and *N. tabacum* plants were grown in a climate-controlled growth chamber (23 ± 1°C, 16:8 hr light:dark cycle).

### Phylogenetic analysis of RDP

To identify potential RDP homologs across insect species, we performed a BLASTp search using the BtRDP protein sequence as a query against the predicted proteomes of 30 insect species. Hits were filtered using an *E*-value cutoff of $10^{-5}$. The retrieved RDP homologs were aligned using MAFFT (v7.310) with default parameters (*Katoh and Standley, 2013*). To improve alignment quality, we removed ambiguously aligned regions using Gblocks with default settings. The optimal amino acid substitution model (JTT+G) was determined using ModelTest-NG (v0.1.6). Phylogenetic relationships were inferred using maximum likelihood analysis in RAxML (v0.9.0) with 1000 bootstrap replicates to assess branch support (*Kozlov et al., 2019*). The final tree was visualized and annotated using iTOL (https://itol.embl.de/).

### Preparation of anti-BtRDP and anti-NtRLP4 serum

The coding sequence of BtRDP (excluding the signal peptide) was PCR-amplified and cloned into the pET-28a vector (Novagen) to generate a C-terminal 6×His-tagged fusion protein. The recombinant plasmid was transformed into *E. coli* Transetta (DE3) competent cells (TransGen Biotech, Beijing, China). Protein expression was induced with 0.5 mM isopropyl-β-D-thiogalactoside (IPTG) at 28°C for 8 hr. Bacterial cells were harvested by centrifugation and lysed via ultrasonication for 30 min. The recombinant BtRDP$^{-sp}$-His protein was purified under native conditions using Ni-NTA agarose (QIAGEN, Hilden, Germany) according to the manufacturer's protocol.

Purified BtRDP$^{-sp}$-His protein was used to immunize rabbits (Hua'an Biotechnology, Hangzhou, China) for custom polyclonal antibody production. The anti-NtRLP4 serum, prepared by immunizing rabbits with peptide GQNSDRPISTKNST, was produced via the custom service of Genscript Biotechnology Company (Nanjing, China). The anti-BtRDP serum was subsequently conjugated to Alexa Fluor 488 NHS Ester (#A20000, Thermo Fisher Scientific) following standard labeling protocols.

The specificity of anti-BtRDP serum was tested by comparing ds*GFP*- and ds*BtRDP*-treated *B. tabaci*. The results showed that ds*BtRDP* significantly reduced the target gene. Similarly, the specificity of anti-NtRLP4 serum was tested by comparing RNAi-EV and RNAi-NtRLP4 plants, which showed a reduced NtRLP4 expression in *RNAi-NtRLP4* plants.

### IHC staining

Dissected *B. tabaci* heads with attached salivary glands and Parafilm-embedded salivary sheaths were fixed in 4% paraformaldehyde (#E672002, Sango Biotechnology, Shanghai) for 30 min. Salivary sheaths adherent to Parafilm were similarly excised and fixed under identical conditions. After fixation, samples were washed three times with PBST (PBS containing 0.1% Tween-20) and blocked with 10% FBS for 2 hr at room temperature. The tissues were then incubated overnight at 4°C with Alexa Fluor 488-conjugated anti-BtRDP serum (1:200 dilution), followed by co-staining with phalloidin–rhodamine (1:500, Thermo Fisher Scientific) for 30 min and 4',6-diamidino-2-phenylindole solution (#ab104139, Abcam, Cambridge, USA) for nuclear visualization. Fluorescent images were acquired using a Leica SP8 confocal microscope (Leica Microsystems) to examine BtRDP localization patterns.

### Protein extraction and western blotting assays

To identify secreted salivary proteins in *N. tabacum* plants, approximately 200 *B. tabaci* adults were confined in a 3-cm diameter feeding cage and allowed to feed on host plants for 24 hr. After feeding, the eggs deposited on the infested tobacco leaves were removed. The leaves showing no visible insect contamination were immediately frozen in liquid nitrogen and ground to a fine powder. The powdered tissue was then homogenized in RIPA Lysis Buffer (#89900, Thermo Fisher Scientific) for

protein extraction. To examine the effects of insect feeding on NtRLP expression, groups of approximately 200 *B. tabaci* were allowed to feed on 4–5 true-leaf stage tobacco plants for 12, 24, or 48 hr, with uninfested plants serving as controls. For silencing/overexpression efficiency tests, leaf samples were collected at 48 hr post-*Agrobacterium* infiltration, along with transgenic tobacco plants at the 4–5 true-leaf stage. For insect tissue analysis, various *B. tabaci* organs were carefully dissected in 1× PBS buffer, including carcasses (20), fat bodies (20), guts (40), salivary glands (60), and ovaries (10), with sample sizes indicated in parentheses. All dissected tissues were transferred to lysis buffer for protein extraction. Total protein concentrations in both insect and plant homogenates were quantified using a BCA Protein Assay Kit (#CW0014S, CwBiotech, Taizhou, China). Equal protein amounts were separated by SDS–PAGE and electro-transferred to PVDF membranes.

Membranes were probed with the following primary antibodies: anti-FLAG (1:10,000, #MA1-91878, Thermo Fisher Scientific), anti-MYC (1:10,000, #MA1-21316, Thermo Fisher Scientific), anti-GFP (1:10,000, #MA5-15256, Thermo Fisher Scientific), anti-NtRLP4 (1:5000, GenScript Biotechnology, Nanjing, China), and anti-BtRDP serum (1:5000, Huaan Biotechnology, Hangzhou, China). Following primary antibody incubation, membranes were treated with HRP-conjugated secondary antibodies: goat anti-mouse IgG (1:10,000, #31430, Thermo Fisher Scientific) or goat anti-rabbit IgG (1:10,000, #31460, Thermo Fisher Scientific). Protein bands were visualized using an AI 680 image analyzer (Amersham Pharmacia Biotech, Buckinghamshire, UK), with Ponceau S staining used to verify equal protein loading.

## qRT-PCR analysis

To prepare tissue-specific samples from *B. tabaci*, adult females were dissected to isolate carcasses (10), fat bodies (10), guts (20), salivary glands (40), and ovaries (10). Additionally, different developmental stages, including eggs (50), nymphs (20), pupae (20), and adult males (20) and females (20), were collected for analysis. The number of insects in each sample is indicated in parentheses. To extract RNA from *N. tabacum* plants, the samples were ground using liquid nitrogen after collection. Total RNA was extracted using the TRIzol Total RNA Isolation Kit (#9109, Takara, Dalian, China) following the manufacturer's instructions. RNA purity and concentration were assessed using a Nano-Drop spectrophotometer (Thermo Fisher Scientific). First-strand cDNA was synthesized from equal amounts of RNA (1 μg) using the HiScript II Q RT SuperMix (#R212-01, Vazyme, Nanjing, China). Gene expression analysis was performed using a Roche LightCycler 480 system (Roche Diagnostics, Mannheim, Germany) with SYBR Green Supermix (#11202ES08, Yeasen, Shanghai, China). The thermal cycling protocol consisted of initial denaturation at 95°C for 5 min and amplification for 40 cycles of 95°C for 10 s and 60°C for 30 s. Primers were designed with Primer Premier v6.0 (*Supplementary file 1D*). *B. tabaci* Tubulin and *N. tabacum* Tubulin served as internal controls for normalization. Relative gene expression was calculated using the $2^{-\Delta\Delta Ct}$ method (*Livak and Schmittgen, 2001*). Three to six independent biological replicates, each repeated twice, were performed.

## RNA interference

The target gene DNA sequences were amplified using primers listed in *Supplementary file 1D* and subsequently cloned into the pClone007 Vector (#TSV-007, Tsingke Biotechnology, Beijing, China). For dsRNA production, PCR-generated DNA templates containing T7 promoter sequences were transcribed in vitro using the T7 High Yield RNA Transcription Kit (#TR101-01, Vazyme Biotech, Nanjing, China). The RNAi experiments were performed according to established methods (*Xu et al., 2015*). Briefly, newly emerged adult whiteflies were anesthetized with carbon dioxide for 5–10 s. Approximately 50 nl of dsRNA solution (2 μg/μl) was microinjected into the mesothorax of each insect using a FemtoJet microinjection system (Eppendorf, Hamburg, Germany). Following injection, the whiteflies were maintained on healthy *N. tabacum* leaves under standard rearing conditions. Only actively moving insects were selected for subsequent analyses. Gene silencing efficiency was evaluated on day 4 post-injection using both qRT-PCR and western blotting approaches.

## Generation of transgenic plants

We generated two transgenic *N. tabacum* lines expressing different BtRDP variants: one containing the full coding sequence (oeBtRDP) and another lacking the signal peptide (oeBtRDP$^{-sp}$), which may affect protein secretion. The target sequences, along with the full-length NtRLP4 coding sequence and

an NtRLP4 hairpin RNAi construct, were each cloned into the pBWA(V)HS vector. The oeBtRDP and oeBtRDP⁻ˢᵖ were fused with C-terminal flag tags, while no tag was fused to oeNtRLP4. All constructs were transformed into *A. tumefaciens* strain GV3101 and used to generate transgenic tobacco plants (K326 background) through *Agrobacterium*-mediated transformation performed by Wuhan Boyuan Biological Company. Transgenic lines were verified by qRT-PCR and western blotting, with empty vector-transformed plants serving as controls, and two independent lines for each construct were selected for further experiments.

## Target gene silencing in plants

The hairpin RNAi construct targeting *NtSOBIR1* was generated by cloning the specific fragment into the pBWA(V)HS plasmid using BsaI and Eco31I restriction enzymes. The recombinant vector was then transformed into *A. tumefaciens* strain GV3101 and subsequently infiltrated into *N. tabacum* leaves using the agroinfiltration method. Following a 48-hr incubation period under standard growth conditions, silencing efficiency was quantitatively evaluated by qRT-PCR analysis.

## Transcriptomic sequencing

To investigate the transcriptional effects of NtRLP4, BtRDP, and BtRDP⁻ˢᵖ in *N. tabacum*, leaf samples from EV, oeRLP4#1, oeBtRDP#1, and oeBtRDP⁻ˢᵖ#1 transgenic plants were homogenized in TRIzol Reagent (#10296018, Invitrogen, Carlsbad, CA, USA) for total RNA extraction following the manufacturer's protocol. Afterward, total RNA was extracted according to the instructions of the manufacturer, and the RNA samples were sent to Novogene Institute (Novogene, Beijing, China) for transcriptomic sequencing. Briefly, poly(A)+ RNA was isolated from 20 μg pooled total RNA using oligo(dT) magnetic beads, followed by fragmentation in divalent cation buffer at 94°C for 5 min. First-strand cDNA synthesis was performed using N6 random primers, with subsequent second-strand synthesis to generate double-stranded cDNA. After end-repair and Illumina adapter ligation, the products were PCR-amplified (15 cycles) and purified using a QIAquick PCR Purification Kit (#28104, QIAGEN, Hilden, Germany). The resulting cDNA libraries were quantified and quality-checked before paired-end sequencing on an Illumina NovaSeq 6000 platform. All raw sequencing data were deposited in the National Genomics Data Center under accession number PRJCA025907.

## Analysis of transcriptomic data

The raw sequencing reads were quality-filtered using Illumina's internal software, and the resulting clean reads from each cDNA library were aligned to the *N. tabacum* reference genome in Sol Genomics Network (https://solgenomics.net/ftp/genomes/Nicotiana_tabacum/edwards_et_al_2017/) using HISAT v2.1.0 (*Kim et al., 2015*). Low-quality alignments were removed using SAMtools (v1.7) with default parameters (*Li et al., 2009*). Transcript abundance was quantified as transcripts per million using Cufflinks (v2.2.1) (*Trapnell et al., 2012*). Differential expression analysis was performed with DESeq2 (v2.2.1) (*Wang et al., 2010*), identifying significantly DEGs using thresholds of |log$_2$(-fold change)| > 1 and adjusted p-value <0.05. PCA of transcriptome profiles was conducted using a custom R script plotPCA (https://github.com/franco-ye/TestRepository/blob/main/PCA_by_deseq2.R, *franco-ye, 2024*). For functional annotation, KEGG pathway enrichment analysis was performed in TBtools (v2.083) using a one-sided hypergeometric test: $p = 1 - \sum_{i=0}^{m-1} \left( \frac{\binom{M}{i}\binom{N-M}{n-i}}{\binom{N}{n}} \right)$ (*Chen et al., 2020*).

In this software, enriched p-values were calculated according to one-sided hypergeometric test: with *N* representing the number of genes with KEGG annotation, *n* standing for the number of DEGs in *N*, *M* indicating the number of genes in each KEGG term, and *m* suggesting the number of DEGs in each KEGG term.

## Insect bioassays

To survivorship analysis, newly emerged *B. tabaci* adults were microinjected with dsRNA and placed on *N. tabacum* leaves. After 24 hr, any dead individuals (potentially due to injection injury) were removed. For each treatment, 20–30 surviving adults were transferred to leaf cages, and mortality was recorded daily for 10 days. Three independent biological replications were performed.

For fecundity analysis, dsRNA-treated virgin females were paired with untreated males in leaf cages. Five such pairs were maintained per cage (*n* = 10–12 cages per treatment). Egg deposition

was quantified daily for 3 consecutive days under controlled conditions. Each male–female pair was considered an experimental replicate.

For the host preference test, detached *N. tabacum* leaves with moist cotton-wrapped petioles were arranged in a 50-cm diameter choice arena with a central release chamber. Forty adult whiteflies were released per chamber, and settling preference was recorded at 3, 6, 12, 24, 36, and 48 hr post-release. The experiment included 10–12 replicates per treatment, with leaf positions randomized between replicates to eliminate positional bias.

## EPG recording and analysis

Feeding behavior of *B. tabaci* was monitored using a GiGA-8d EPG amplifier (Wageningen Agricultural University; 10 TΩ input resistance, <1 pA input bias current) with 50× gain and ±5 V output range. Prior to recording, dsRNA-treated whiteflies were starved for 12 hr on moist filter paper, briefly anesthetized with $CO_2$ (10 s), and connected to the amplifier via a gold wire electrode (20 μm diameter × 5 cm length) attached to the pronotum using conductive silver glue. Plant electrodes consisted of copper wires (2 mm diameter × 10 cm length) inserted into the soil of potted *N. tabacum*. Recordings were conducted for 8 hr in a Faraday cage (120 × 75 × 67 cm; Dianjiang, Shanghai), with only insects surviving the full duration retained for analysis.

Recorded waveforms were processed using PROBE 3.4 software (Wageningen Agricultural University) and categorized into four distinct feeding phases: nonpenetration (np), pathway duration (C), phloem salivation (E1), and phloem ingestion (E2) as defined in a previous study (*Lu et al., 2021*). Each treatment included ≥10 biological replicates, with ds*GFP*-injected and untreated whiteflies serving as controls.

## *Agrobacterium*-mediated plant transformation in *N. tabacum* and *N. benthamiana*

The recombinant expression vectors were introduced into *A. tumefaciens* strain GV3101 via heat shock transformation. Transformed cells were selected on LB agar plates supplemented with 50 μg/ml kanamycin and 10 μg/ml rifampicin, followed by incubation at 28°C for 60 hr. Positive colonies were inoculated into liquid LB medium with the same antibiotics and grown to log phase. Bacterial cells were then pelleted by centrifugation at 2400 × *g* for 2 min and resuspended in induction buffer (10 mM $MgCl_2$, 10 mM MES [pH 5.6], 200 μM acetosyringone) adjusted to specific $OD_{600}$ values depending on the experiment.

For standard experiments, bacterial suspensions were normalized to $OD_{600} = 1.0$, while degradation assays used three concentrations ($OD_{600} = 0.05$, 0.3, and 1.0). Equal volumes of selected bacterial suspensions were mixed and infiltrated into leaves of 4- to 5-week-old *N. tabacum* and *N. benthamiana* plants using a needleless syringe. Infiltrated plants were maintained under controlled conditions for subsequent analysis.

## Diaminobenzidine staining

To detect $H_2O_2$ accumulation in *N. tabacum* leaves, leaf samples were immersed in 1 mg/ml 3,3′-diaminobenzidine tetrahydrochloride (DAB-HCl, #D8001, Sigma-Aldrich, St. Louis, MO, USA) solution (pH 3.8) for 6 hr at room temperature, allowing $H_2O_2$-dependent brown polymerization to occur. Following incubation, the DAB solution was replaced with 100% ethanol for chlorophyll removal, and samples were decolorized overnight at 65°C. The destained leaves were then photographed under standardized conditions using a Canon EOS 80D digital camera (Canon Inc, Tokyo, Japan).

## Interaction assays between two proteins

In the Y2H screening assay, the coding sequence of BtRDP$^{-sp}$ was cloned into the pGBKT7 bait vector (Clontech, USA), while a *N. benthamiana* cDNA library was constructed in the pGADT7 prey vector (Biogene Biotech, Shanghai, China). These recombinant vectors were co-transformed into yeast strain Y2HGold and plated on QDO medium (SD/-Ade/-His/-Leu/-Trp, #630428, Takara) for 3 days at 30°C. Positive colonies were cultured in QDO liquid medium, and plasmids were extracted using a yeast DNA kit (#DP112-02, TIANGEN, Beijing, China) before transformation into *E. coli* DH5α (#TSC-C14, Tsingke) for sequencing (YouKang Biotech, China) to identify interacting partners.

In the pairwise Y2H verification assay, BtRDP$^{-sp}$ and NlSP104$^{-sp}$, and mutant variants of NtRLP4/ SlRLP4/OsRLP4 were cloned into either pGBKT7 or pGADT7 vectors using primers from *Supplementary file 1D*. Given that Y2H is generally difficult to identify membrane receptors, the truncated versions of NtRLP4/SlRLP4/OsRLP4 lacking the signal peptide and transmembrane domains were used. Yeast co-transformants were first selected on DDO medium (SD/-Leu/-Trp) (#630417, Takara) for 3 days at 30°C, then spotted onto QDO medium to confirm interactions through growth after another 3 days of incubation.

In the Co-IP assays, full-length sequences of *BtRDP*, *NlSP104*, *NtSOBIR1*, *NtRLP4*, *SlRLP4*, and *OsRLP4* were cloned into lic-myc or lic-flag vectors for C-terminal tagging, with BtFTSP-flag serving as a negative control. Five-week-old *N. benthamiana* leaves were agroinfiltrated with these constructs or collected from oeBtRDP transgenic plants. Total proteins were extracted from 1 g frozen tissue in IP lysis buffer (#87788, Thermo Scientific) with protease inhibitors (#56079200, Roche, Switzerland), followed by centrifugation at 1000 × *g* for 20 min. Cleared lysates were incubated with anti-flag beads (#L00432-1, GenScript, Nanjing, China) for 4 hr at 4°C, washed with PBS, and eluted in 2× SDS–PAGE buffer (500 mM Tris-HCl, pH = 6.8, 50% glycerin, 10% SDS, 1% bromophenol blue, and 2% β-mercaptoethanol) for western blotting assay.

In the BiFC assay, BtRDP$^{-sp}$, BtFTSP$^{-sp}$, NtRLP4, and NtCf9-like were cloned into the pCV-cYFP or pCV-nYFP vectors, respectively. After transforming *A. tumefaciens* GV3101, pairwise combinations were co-infiltrated into *N. benthamiana* leaves. Following 36- to 48-hr incubation in a growth chamber, reconstituted YFP fluorescence was visualized using a Leica SP8 confocal microscope with standard filter sets for YFP detection.

## RLP degradation by salivary effectors among different herbivorous insects

To examine salivary protein effects on RLPs, we co-infiltrated *A. tumefaciens* GV3101 strains carrying various constructs into *N. benthamiana* leaves. The recombinant vectors expressed C-terminally tagged proteins: BtRDP-flag, NlSP104-flag, GUS-flag, NlSP7-flag, and myc-tagged RLPs (NtRLP4, SlRLP4, OsRLP4, SlCf9, and NtCf9). Bacterial suspensions were mixed at predetermined $OD_{600}$ ratios before infiltration, with leaf samples collected 48 hr post-infiltration for subsequent analysis.

For confocal microscopy studies, *N. benthamiana* leaves were co-infiltrated with *A. tumefaciens* carrying: (1) NtRLP4-GFP plus BtRDP-mCherry, (2) NtRLP4-GFP plus RFP-mCherry, or (3) NtRLP4-GFP plus BtFTSP-mCherry. The samples were imaged by confocal microscopy at 48 hr post-injection. The same parameters were used in photographing.

This study also conducted a treatment on *N. benthamiana* plants that were overexpressing NtRLP4-myc using purified proteins. Briefly, the BtRDP$^{-sp}$ and GFP coding sequences were cloned into pET-28a for 6×His-tagged expression in *E. coli* Transetta (TransGen Biotech). Protein expression was induced with 0.5 mM IPTG at 25°C for 8 hr, followed by purification using Ni-NTA agarose and buffer exchange into PBS using a 3-kDa molecular-weight cutoff Amicon Ultra-4 Centrifugal Filter Device (Millipore, MA, USA). Different concentrations of purified proteins were infiltrated into leaves of *N. benthamiana* plants stably expressing NtRLP4-myc, with samples collected 24 hr post-treatment.

To investigate degradation pathways, we infiltrated *N. benthamiana* leaves with: 50 µM MG132 (#M7449, Sigma-Aldrich, Steinheim, Germany), 5 µM 3-MA (#M9281, Sigma-Aldrich), or 10 nM BAF (#BML-CM110-0100, Enzo Life Sciences, Farmingdale, USA) at 24 hr post-agroinfiltration. The samples were incubated for an additional 24 hr. Protein levels were detected using western blotting assay.

## In planta ubiquitination assay

Ubiquitination detection in *N. benthamiana* was performed according to the methods previously reported (*Wang et al., 2022*). The recombinant vector for transient expression of the HA-fused ubiquitin protein was kindly provided by Dr. Chao Zheng (Ningbo University). Myc-tagged proteins were co-expressed with HA-UBQ by agroinfiltration, and 50 µM MG132 was injected into the *N. benthamiana* leaves 1 day post *A. tumefaciens* infiltration. Immunoprecipitation was performed using

anti-myc beads (ytma-20, ChromoTek, Planegg-Martinsried, Germany). Ubiquitination signals were detected by immunoblot with anti-HA antibody (HT301, TransGen Biotech, Beijing, China).

## Scanning electron microscopy

*B. tabaci* adults were allowed to feed on artificial diet solutions for 24 hr. Parafilm sections containing salivary sheaths were carefully excised and rinsed with 1× PBS (pH 7.4) to remove residual diet. The samples were mounted on aluminum stubs, desiccated under vacuum, and sputter-coated with gold before imaging with a TM4000 II Plus SEM (Hitachi, Tokyo, Japan). Salivary sheath length (from apex to base) was measured, with 20 sheaths analyzed per treatment to ensure statistical reliability.

## Statistical analysis

The log-rank test (SPSS Statistics 19, Chicago, IL, USA) was used to determine the statistical significance of survival distributions. Two-tailed unpaired Student's $t$-test (comparisons between two groups) or one-way ANOVA test followed by Tukey's multiple comparisons test (comparisons among three groups) was used to analyze the results of qRT-PCR, fecundity analysis, and host choice analysis. The EPG data were first checked for the normality and homogeneity of variance, and those not fitting a normal distribution were $\log_{10}$ transformed before Student's $t$-test, as described in previous studies (*Lu et al., 2021*). Statistical tests, including Student's $t$-test and one-way ANOVA, and data visualization were conducted in GraphPad Prism 9.

## Acknowledgements

This project has received funding from the National Natural Science Foundation of China (32422075: HJH; U23A6006: JPC and CXZ), National Key Research and Development Program of China (2021YFD1401100: HJH and CXZ), and Natural Science Foundation of Zhejiang Province (LDQ24C140001: HJH).

## Additional information

### Funding

| Funder | Grant reference number | Author |
| --- | --- | --- |
| National Natural Science Foundation of China | 32422075 | Hai-Jian Huang |
| National Natural Science Foundation of China | U23A6006 | Jian-Ping Chen Chuan-Xi Zhang |
| National Key Research and Development Program of China | 2021YFD1401100 | Hai-Jian Huang Chuan-Xi Zhang |
| Natural Science Foundation of Zhejiang Province | LDQ24C140001 | Hai-Jian Huang |

The funders had no role in study design, data collection, and interpretation, or the decision to submit the work for publication.

### Author contributions

Xin Wang, Data curation, Formal analysis, Validation, Investigation, Visualization, Writing – review and editing; Jia-Bao Lu, Yi-Zhe Wang, Investigation; Xu-Hong Zhou, Resources; Jian-Ping Chen, Conceptualization, Funding acquisition; Chuan-Xi Zhang, Conceptualization, Supervision; Jun-Min Li, Conceptualization; Hai-Jian Huang, Conceptualization, Writing – original draft, Writing – review and editing

### Author ORCIDs

Chuan-Xi Zhang ⓘ https://orcid.org/0000-0002-7784-1188
Hai-Jian Huang ⓘ https://orcid.org/0000-0002-0968-4520

Reviewer #1 (Public review): https://doi.org/10.7554/eLife.108737.4.sa1
Reviewer #2 (Public review): https://doi.org/10.7554/eLife.108737.4.sa2
Reviewer #3 (Public review): https://doi.org/10.7554/eLife.108737.4.sa3
Author response https://doi.org/10.7554/eLife.108737.4.sa4

## Additional files

### Supplementary files

Supplementary file 1. Sequences, accessions, and expression patterns of associated genes in this study. (A) Identification of RDP and SP101 homologs in insect species. (B) Proteins from a *Nicotiana benthamiana* cDNA library screened by yeast two-hybrid using BtRDP as a bait. (C) Differentially expressed genes between empty vector (EV) and oeRLP#1 transgenic plant. (D) Primers used in this study.

MDAR checklist

### Data availability

All data is included in the manuscript and/or supporting information. The transcriptomic data have been submitted to the National Genomics Data Center under accession number PRJCA025907 [https://ngdc.cncb.ac.cn/bioproject/browse/PRJCA025907]. Source data files have been provided for all figures.

The following dataset was generated:

| Author(s) | Year | Dataset title | Dataset URL | Database and Identifier |
|-----------|------|---------------|-------------|-------------------------|
| Huang H-J | 2026 | Transcriptome analysis of Nicotiana tabacum overexpressing NtRLP4 | https://ngdc.cncb.ac.cn/bioproject/browse/PRJCA025907 | National Genomics Data Center, PRJCA025907 |

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
