## [Editor Report · eLife Assessment]

This study provides an **important** contribution by showing that whiteflies and planthoppers use salivary effectors to suppress plant immunity through the receptor-like protein RLP4, suggesting convergent evolution in these insect lineages. The topic is of clear interest for understanding plant-insect interactions and offers ideas that could stimulate further research in the field. The authors provide **convincing** evidence for the functional roles of the salivary effectors.

---

## [Referee Report · Reviewer #1 (Public review)]

Summary:

This manuscript investigates how herbivorous insects, specifically whiteflies and planthoppers, utilize salivary effectors to overcome plant immunity by targeting the RLP4 receptor.

Strengths:

The authors present a strong case for the independent evolution of these effectors and provide compelling evidence for their functional roles.

Comments on revisions:

The authors have addressed all my concerns.

---

## [Referee Report · Reviewer #2 (Public review)]

Summary:

The authors tested an interesting hypothesis that white flies and planthoppers independently evolved salivary proteins to dampen plant immunity by targeting a receptor like protein. Unlike previously reported receptor like proteins with large ligand-binding domains, the NtRLP4 here has a malectin LRR domain. Interestingly, it also associates with the adaptor SOBIR1. While the function of this protein remains to be further explored, the authors provide strong evidence showing it's the target of salivary proteins as the insects' survival strategy.

The authors have nicely addressed the questions I raised.

I noticed two small points the authors may modify:

- Line 16: delete "on"

- Line 185: Replace "is resistant to B. tabaci infestation" with "confers resistance against B. tabaci".

---

## [Referee Report · Reviewer #3 (Public review)]

Summary:

In this study, Wang et al., investigates how herbivorous insects overcome plant receptor-mediated immunity by targeting plant receptor-like proteins. The authors identify two independently evolved salivary effectors, BtRDP in whiteflies and NlSP694 in brown planthoppers, that promote the degradation of plant RLP4 through the ubiquitin-dependent proteasome pathway.

Strengths:

This work highlights a convergent evolutionary strategy in distinct insect lineages and advances our understanding of insect-plant coevolution at the molecular level.

Comments on revisions:

The authors have satisfactorily addressed all the issues I raised.

---

## [Author Response]

The following is the authors’ response to the previous reviews

**Public Reviews:**

**Reviewer #1 (Public review):**
Summary:This manuscript investigates how herbivorous insects, specifically whiteflies and planthoppers, utilize salivary effectors to overcome plant immunity by targeting the RLP4 receptor.

Thank you for your comments.

Strengths:The authors present a strong case for the independent evolution of these effectors and provide compelling evidence for their functional roles.

Thank you for your help in improving our manuscript

**Reviewer #2 (Public review):**
Summary:The authors tested an interesting hypothesis that white flies and planthoppers independently evolved salivary proteins to dampen plant immunity by targeting a receptor-like protein. Unlike previously reported receptor-like proteins with large ligand-binding domains, the NtRLP4 here has a malectin LRR domain. Interestingly, it also associates with the adaptor SOBIR1. While the function of this protein remains to be further explored, the authors provide strong evidence showing it's the target of salivary proteins as the insects' survival strategy.

Thank you for your comments.

Major points:The authors mixed the concepts of LRR-RLPs with malectin LRR-RLPs. These are two different type of receptors. While LRR-RLPs are well studied, little is known about malectin LRR-RLPs. The authors should not simply apply the mode of function of LRR-RLPs to RLP4 which is a malectin LRR-RLP. In addition, LRR-RLPs that function as ligand-binding receptors typically possess >20 LRRs, whereas RLP4 in this work has a rather small ectodomain. It remains unclear whether it will function as a PRR. I can't agree with the author's logic of testing uninfested plants for proving a PRR's function. The function of a pattern recognition receptor depends on perceiving the corresponding ligand. As shown by the data provided, RLP4-OE plants have altered transcriptional profile indicating activated defense, suggesting it's unlikely a PRR. An alternative explanation is needed. More work on BAK1 will also help to clarify the ideas proposed by the authors.

We sincerely thank the reviewer for the insightful and constructive comments, which have helped us critically re-evaluate our interpretation of RLP4 function. In the revised manuscript, we have addressed this important point by adding a detailed discussion of an alternative explanation for RLP4’s role in plant defense. Specifically, we now explicitly distinguish between classical LRR-RLPs and malectin-domain-containing RLPs, and we acknowledge that RLP4 may not function as a canonical PRR. We also discuss the structural features of RLP4, including its malectin-like domain and relatively small LRR region, and the observation that NtRLP4 overexpression lines exhibit altered transcriptional profiles even in the absence of insect infestation. Based on these lines of evidence, we propose that RLP4 may instead act as a regulatory component within plant immune signaling networks, modulating defense outputs rather than functioning as a direct ligand receptor. The revised discussion now reads as follows: “Together, this study reveals that suppressing PRR-mediated plant immunity may be a conserved strategy employed by herbivorous insects for successful feeding. We demonstrate that whiteflies and planthoppers have independently evolved salivary effectors that facilitate the ubiquitin-dependent degradation of defensive RLP4 in host plants, thereby dampen RLP4-mediated plant immunity (Fig. 6). Nevertheless, the precise mechanism by which RLP4 contributes to plant defense warrants further consideration. While it may function as a canonical PRR that perceives insect-derived molecular patterns, several lines of evidence point to an alternative interpretation. Structurally, RLP4 differs from classical LRR-RLP: it contains a malectin-like domain and a relatively small LRR domain, contrasting with typical LRR-RLPs that often possess large LRRs dedicated to ligand binding. Functionally, NtRLP4 overexpression lines exhibit significantly altered transcriptional profiles and dysregulated SA/JA pathways even in the absence of insect infestation, a phenotype inconsistent with canonical PRRs, which typically remain quiescent until ligand perception. These findings point to an alternative explanation: rather than functioning as a classical PRR that recognizes insect-derived molecules, RLP4 may act as a regulatory component within plant immune signaling networks. Elucidating the precise mechanism of RLP4 in conferring plant defense against herbivorous insects will therefore be an important focus of future research” in Line 392-407.

**Reviewer #3 (Public review):**
Summary:In this study, Wang et al., investigate how herbivorous insects overcome plant receptor-mediated immunity by targeting plant receptor-like proteins. The authors identify two independently evolved salivary effectors, BtRDP in whiteflies and NlSP694 in brown planthoppers, that promote the degradation of plant RLP4 through the ubiquitin-dependent proteasome pathway. NtRLP4 from tobacco and OsRLP4 from rice are shown to confer resistance against herbivores by activating defense signaling, while BtRDP and NlSP694 suppress these defenses by destabilizing RLP4 proteins.

Thank you for your comments.

Strengths:This work highlights a convergent evolutionary strategy in distinct insect lineages and advances our understanding of insect-plant coevolution at the molecular level.Two minor comments:In line 140, yeast two-hybrid (Y2H) was used to screen for interacting proteins in plants. However, it is generally difficult to identify membrane receptors using Y2H. Please provide more methodological details to justify this approach, or alternatively, include a discussion explaining this.

Thank you for pointing this out. It is true that Y2H is generally difficult to identify membrane receptors. To address this limitation, we used truncated versions of RLP4s lacking the signal peptide and transmembrane domains in point-to-point Y2H assays. In addition, the interactions between BtRDP and RLP4s were further validated by Co-IP and BiFC experiments. In the revised manuscript, we have clarified this methodological detail as follows: “Given that Y2H is generally difficult to identify membrane receptors, the truncated versions of NtRLP4/SlRLP4/OsRLP4 lacking the signal peptide and transmembrane domains were used” in Linr 636-638.

In Figure S12C, the interaction between the two proteins appears to be present in the nucleus as well. Please provide a possible explanation for this observation.

Thank you for pointing this out. During revision, we further examined the subcellular localization of NtRLP4 and found that NtRLP4-GFP could also be detected in the nucleus when expressed alone (Fig. S18), suggesting that NtRLP4 may have additional functions beyond serving as a cell surface pattern recognition receptor. In the revised manuscript, we discussed that NtRLP4 might play other roles in addition to PRRs in the discussion section as follow: “Together, this study reveals that suppressing PRR-mediated plant immunity may be a conserved strategy employed by herbivorous insects for successful feeding. We demonstrate that whiteflies and planthoppers have independently evolved salivary effectors that facilitate the ubiquitin-dependent degradation of defensive RLP4 in host plants, thereby dampen RLP4-mediated plant immunity (Fig. 6). Nevertheless, the precise mechanism by which RLP4 contributes to plant defense warrants further consideration. While it may function as a canonical PRR that perceives insect-derived molecular patterns, several lines of evidence point to an alternative interpretation. Structurally, RLP4 differs from classical LRR-RLP: it contains a malectin-like domain and a relatively small LRR domain, contrasting with typical LRR-RLPs that often possess large LRRs dedicated to ligand binding. Functionally, NtRLP4 overexpression lines exhibit significantly altered transcriptional profiles and dysregulated SA/JA pathways even in the absence of insect infestation, a phenotype inconsistent with canonical PRRs, which typically remain quiescent until ligand perception. These findings point to an alternative explanation: rather than functioning as a classical PRR that recognizes insect-derived molecules, RLP4 may act as a regulatory component within plant immune signaling networks. Elucidating the precise mechanism of RLP4 in conferring plant defense against herbivorous insects will therefore be an important focus of future research” in Line 392-407.

**Recommendations for the authors:**

**Reviewer #1 (Recommendations for the authors):**
The authors have addressed all my concerns.

Thank you for your help in improving our manuscript

**Reviewer #2 (Recommendations for the authors):**
This work is quite interesting. It's not necessary to prove RLP4 as a PRR to show the merit of this discovery. The current logic is forced and thus the conclusion not convincing. Finding an alternative explanation will be more helpful.

Thank you for your valuable suggestions. In the revised version, we discussed the alternative explanation as follow: “Together, this study reveals that suppressing PRR-mediated plant immunity may be a conserved strategy employed by herbivorous insects for successful feeding. We demonstrate that whiteflies and planthoppers have independently evolved salivary effectors that facilitate the ubiquitin-dependent degradation of defensive RLP4 in host plants, thereby dampen RLP4-mediated plant immunity (Fig. 6). Nevertheless, the precise mechanism by which RLP4 contributes to plant defense warrants further consideration. While it may function as a canonical PRR that perceives insect-derived molecular patterns, several lines of evidence point to an alternative interpretation. Structurally, RLP4 differs from classical LRR-RLP: it contains a malectin-like domain and a relatively small LRR domain, contrasting with typical LRR-RLPs that often possess large LRRs dedicated to ligand binding. Functionally, NtRLP4 overexpression lines exhibit significantly altered transcriptional profiles and dysregulated SA/JA pathways even in the absence of insect infestation, a phenotype inconsistent with canonical PRRs, which typically remain quiescent until ligand perception. These findings point to an alternative explanation: rather than functioning as a classical PRR that recognizes insect-derived molecules, RLP4 may act as a regulatory component within plant immune signaling networks. Elucidating the precise mechanism of RLP4 in conferring plant defense against herbivorous insects will therefore be an important focus of future research” in Line 392-407.

Inappropriate descriptions still exist at multiple places across the manuscript and damages the merit of this work. I highly recommend the authors to consult an expert in plant PRR research for proof reading. The language editing service the authors used only provided limited help in this case. Here are a few examples:

We sincerely thank the reviewer for the critical and constructive comments. We agree that precise language is essential for conveying scientific findings. In the revised version, we have refined the text with the help of colleagues who have expertise in plant immunity, aiming to ensure the descriptions are as precise and professional as possible.

Line 16: Using "depend" ignores the fact that many biotic invaders are recognized by NLRs. The authors can simply use the word "use" or "utilize".

Thank you for your suggestion. We corrected it in the revised version.

Line 20:"target defensive RLP4, therefor minimizing the plant immunity" is a strange saying. "dampen RLP4-mediated plant immunity"will be better.

Thank you for your suggestion. We corrected it in the revised version.

Line 49: as far as I know, only LRR-RLPs use SOBIR1 as adaptor. The authors should introduce this specific point. The mode of action of other type of LRR-RLPs are less clear.

Thank you for your suggestion. In the revised version, we re-introduce this as follow: “As RLPs lack the intracellular signaling domains, they are anticipated to associate with adaptor kinases to form the bimolecular receptor kinases. For example, suppressor of BAK1-interacting receptor-like kinase 1 (SOBIR1) is reported to act as a common adaptor for most, if not all, of the leucine-rich repeat RLP (LRR-RLP)” in Line 48-52, “The receptor-like kinase SOBIR1, which contained a kinase domain, has been widely reported to be required for the function of LRR-RLPs in the innate immunity. However, whether SOBIR1 interacted with malectin-LRR RLP remains largely unknown” in Line 170-173.

Line 67: There are quite a few publications showing that insect salivary proteins dampen plant immunity.

Sorry for the inaccurate description. We agree that an accumulated literature describes the suppression of plant immunity by insect salivary proteins. However, the specific molecular mechanism by which these proteins target plant PRRs is still poorly understood. In the revised version, we specified that “it remains largely unknown how insects cope with plant PRRs” in Line 68-69.

Line 149: I don't understand what "point-to-point Y2H" is.

Thank you for your comment. We agree that the term "pairwise Y2H" is more commonly used in the literature than "point-to-point Y2H." To avoid any confusion and to align with standard terminology, we have replaced "point-to-point Y2H" with "pairwise Y2H" throughout the revised manuscript.

Line 179: Replace with "NtRLP4 and NtSOBIR1 confers resistance to B. tabaci". You don't say a protein is resistant to a insect infestation. The same applies for Line 209-210.

Thank you for your suggestion. We corrected it in the revised version.

Minor points:Line 91-92: Lengthy text for simple results.Line 98: "which was significantly different from the actin or ribosomal 18S rRNA" can be deleted. It's self-evident that actin and 18S rRNA are controls. The same applies to Line 101.Line 130: unnecessary sentence, delete.The use of verb forms needs further correction.

Thank you for your valuable suggestion. In the revised manuscript, we have revised the text accordingly. We truly appreciate your help in improving our manuscript.